# Graph Neural Network Based Action Ranking for Planning

**Rajesh Mangannavar**
Oregon State University
Corvallis, OR 97330, USA
`mangannr@oregonstate.edu`

**Stefan Lee**
Oregon State University
Corvallis, OR 97330, USA
`leestef@oregonstate.edu`

**Alan Fern**
Oregon State University
Corvallis, OR 97330, USA
`alan.fern@oregonstate.edu`

**Prasad Tadepalli**
Oregon State University
Corvallis, OR 97330, USA
`prasad.tadepalli@oregonstate.edu`

## Abstract

We propose a novel approach to learn relational policies for classical planning based on learning to rank actions. We introduce a new graph representation that explicitly captures action information and propose a Graph Neural Network (GNN) architecture augmented with Gated Recurrent Units (GRUs) to learn action rankings. Unlike value-function based approaches that must learn a globally consistent function, our action ranking method only needs to learn locally consistent ranking. Our model is trained on data generated from small problem instances that are easily solved by planners and is applied to significantly larger instances where planning is computationally prohibitive. Experimental results across standard planning benchmarks demonstrate that our action-ranking approach not only achieves better generalization to larger problems than those used in training but also outperforms multiple baselines (value function and action ranking) methods in terms of success rate and plan quality.

## 1 Introduction

Classical planning tackles the problem of finding action sequences to achieve goals in deterministic environments. While traditional planners can solve small problems optimally using search and heuristics, they often struggle with scalability in complex problems [2, 17]. This has motivated research into learning general relational policies from small solved problems, which can then be applied to significantly larger instances [23]. The key insight behind learning-based planning is that optimal solutions to small problems often reveal patterns that generalize to bigger problems. For example, in Blocks World, solutions to 4 block problems can teach a policy to stack blocks bottom-up that generalizes to problems with 20 or more blocks. The advantage of relational learning is its ability to capture compositional structure, which in turn enable strong generalization [30, 6] [4, 9].

Learning approaches in automated planning have traditionally focused on learning heuristic functions to guide search algorithms. These methods typically learn a value function that estimates the distance to the goal and integrate it within classical search algorithms like A* or greedy best-first search. The learned heuristics help focus the search but still require explicit search during plan execution.

Recently, several neural approaches have been proposed in the planning domain. Some methods learn value functions to guide search processes [22, 1], while others learn value functions that induce greedy policies by selecting actions leading to states with minimum estimated cost-to-go [24, 23].

39th Conference on Neural Information Processing Systems (NeurIPS 2025).

While these approaches show promise, they all face a fundamental challenge: value functions are not only complex and challenging to learn, but since optimal planning is NP-hard in most domains [5, 11], there is no reason to think that optimal value functions generalize to larger problem sizes. In our work, instead of trying to learn optimal value functions we focus on learning a policy that ranks good actions higher in any given state.

Learning to rank approaches have shown promise in planning domains but have primarily focused on ranking states rather than actions. [10] pioneered this direction by using RankSVM to learn state rankings using hand-crafted features. More recently, [3], and [13] demonstrated that learning to rank states can be more effective than learning precise heuristic values, as the relative ordering of states is sufficient for guiding search. However, these approaches still fundamentally rely on search during execution, inheriting significant computational overhead. The other drawback is that, while ranking states is easier than learning the exact value function, they are still trying to learn a globally consistent state ranking function which is challenging. We overcome this issue in our work by learning only local ranking over actions in any given state instead of learning a ranking among all possible states. Unlike other action ranking approaches such as [19], [16], [8], [15], and [25], we explicitly represent action information in our input state representation.

Our work is inspired by the effectiveness of graph neural networks (GNNs) [20] to represent and learn general relational policies such as [24, 25, 26, 30] where GNNs have been shown to be able to learn well over state representations for planning problems [18, 1]. GNNs are especially suited for these types of problems as they can handle inputs of varied sizes and can learn from graphs of small size and generalize to larger graphs [12]. This is the property we are looking to exploit - train a system on small sized planning problems where it is easy to run planners and collect data and use this learned model to solve larger problems that planners are too slow to solve while simplifying the learning problem by only learning local action rankings.

To achieve this, we introduce a novel architecture GABAR (GrAph neural network Based Action Ranking), which directly learns to rank actions rather than estimating value functions. GABAR consists of three key components: (1) an action-centric graph representation of state that explicitly captures how objects participate in actions (2) a GNN encoder that processes this rich representation, and (3) a GRU-based decoder that sequentially constructs parameterized actions. Our key insight is that ranking actions that are applicable in the same state often turns out to be easier and more generalizable than ranking states by their distances to the goals. Through experiments on standard benchmarks, we show that (a) GABAR achieves generalization to significantly larger problems than those used for training, and (b) it does markedly better on the larger problems when compared to value-function-based methods or other methods that do not include action information in the graph.

## 2   Problem Setup

Classical planning deals with finding a sequence of actions that transform an initial state into a goal state. A classical planning problem is represented as a pair $P = \langle D, I \rangle$, where $D$ represents a first-order domain and $I$ contains instance-specific information. The domain $D$ consists of a set of predicate symbols $\mathcal{P}$ with associated arities and a set of action schemas $\mathcal{A}$. Each action schema $a \in \mathcal{A}$ is defined by a set of parameters $\Delta(a)$ representing variables that can be instantiated, preconditions $pre(a)$, add effects $add(a)$, and delete effects $del(a)$. The instance information $I$ is a tuple $\langle O, s_0, G \rangle$ where $O$ is a finite set of objects, $s_0$ is the initial state represented as a set of ground atoms $p(o_1, ..., o_k)$ where $p \in \mathcal{P}$ and $o_i \in O$, and $G$ is the goal condition also represented as a set of ground atoms.

A state $s$ is a set of ground atoms that are true in that state. An action schema can be grounded by substituting its parameters with objects from $O$. A ground action $a$ is applicable in state $s$ if $pre(a) \subseteq s$, and results in successor state $s' = (s \setminus del(a)) \cup add(a)$. A solution or plan is a sequence of applicable ground actions that transform the initial state $s_0$ into a state satisfying the goal condition $G$. A relational policy maps a problem state to an action. The current paper addresses the following problem. Given a domain $D$ and a set of training instances of different sizes and their solutions, learn a relational policy that leads to efficient solutions for larger test instances from the same domain.

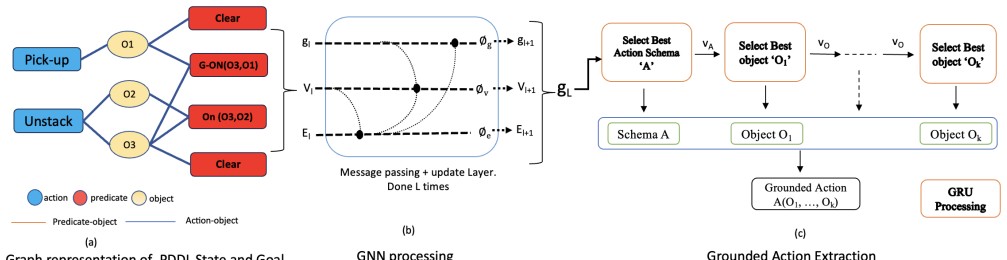

Figure 1: GABAR's architecture for action extraction. (a) Graph representation: The input PDDL problem is converted into a graph with four types of nodes (predicate, object, action schema, and global) connected by predicate-object and action-object edges that encode state and grounded action information. (b) GNN encoder: Processes the graph through $L$ rounds of message passing where edge, node, and global representations are sequentially updated (c) Action decoder: Uses the final global embedding to construct a grounded action through a GRU-based decoder sequentially - first selecting an action schema, then iteratively choosing objects for each parameter position until a complete grounded action is formed.

## 3 GNN Based Action Ranking

Learning general policies for classical planning domains requires learning to select appropriate actions effectively to variable-sized states and generalize across problem instances. This section details each component of GABAR, the graph representation, the GNN encoder and the GRU decoder in detail and explains how they work together to enable effective action selection.

### 3.1 System Overview

Given a planning instance, **I** in PDDL format [7], along with the set of ground actions, GABAR operates by first converting this to a graph structure that makes explicit the relationships between objects, predicates, and potential actions. This graph is then processed through our neural architecture to rank actions(as described in Fig 1). Then, the highest-ranked applicable action is executed to reach the next state **I'**, and the process repeats. To ensure that the execution terminates, the system maintains a history of visited states and avoids actions that would lead to previously visited states. The execution continues until either a goal state is reached or a state is reached with no unvisited successor states, or the maximum execution length (1000 in our experiments) is exceeded.

### 3.2 Graph Representation

We introduce a novel graph representation (shown in Fig 2) for classical planning tasks that captures the structural relationships between objects, predicates, actions, and the semantic information needed to effectively learn action ranking. Our representation $G = (V, E, X, R)$ consists of a set of nodes $V$, edges $E$, node features $X$ and edge features $R$.

The node set $V = O \cup P \cup A \cup \{g\}$ where $O$, $P$,and $A$ represent sets of domain objects, grounded predicates (predicates instantiated in current and goal states), and action schemas, respectively, and $g$ is a global node that aggregates graph-level information. The edge set $E = E_{\text{pred}} \cup E_{\text{act}}$, where $E_{\text{pred}}$ is the set of edges between predicates and their argument objects and similarly $E_{\text{act}}$ is the set of edges between action schemas and their argument objects.

**Node Features:** The node feature function $X : V \rightarrow \mathbb{R}^d$ maps each node to a feature vector that encodes type and semantic information. The feature vector is constructed by concatenating several one-hot encoded segments. For any node $v \in V$, $X(v) = [X_{\text{type}}(v) \parallel X_{\text{act}}(v) \parallel X_{\text{pred}}(v) \parallel X_{\text{obj}}(v)]$, where,

- $X_{\text{type}} \in \{0, 1\}^3$: One-hot encoding of node type (object, predicate, or action)

- $X_{\text{act}} \in \{0, 1\}^{|A|}$: One-hot encoding of action type (if $v$ is an action node)

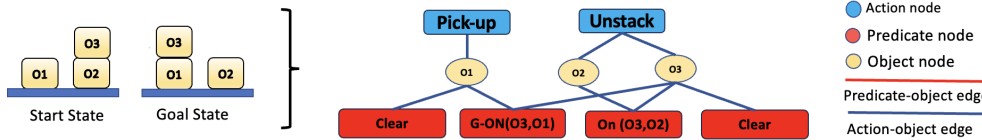

Figure 2: Example graph construction for a simplified blocksworld problem with only *on* and *clear* predicates. The left side shows the starting and goal states. The $O$ nodes are the object nodes. In the start state, $O3$ is on $O2$ which is on the table, and $O1$ is on the table. In the goal state, $O3$ is on $O1$ which is on the table, and $O2$ is on the table. The right side shows the constructed graph with action nodes (blue), object nodes (yellow), and predicate nodes (red). The "Pick-up" action connects to object $O1$, while the "Unstack" action connects to objects $O2$ and $O3$. Predicate nodes show the current state ("Clear" for $O1$ and $O3$, "On (O3,O2)") and goal state ("G-ON(O3,O1)"). Blue edges represent action-object connections, while red edges represent predicate-object relationships.

- $X_{\text{pred}} \in \{0,1\}^{2|P|}$: Encoding for predicates, where first $|P|$ bits indicate predicate type and next $|P|$ bits indicate goal predicates (if $v$ is a predicate node)

- $X_{\text{obj}} \in \{0,1\}^{|T|}$: One-hot encoding of the object type (if $v$ is an object node). $T$ is the set of object types

**Edge Features:** The edge feature function $R : E \to \mathbb{R}^k$ maps each edge to a feature vector that encodes the type of edge and the information about the role. For any edge $e \in E$, $R(e) = [R_{\text{type}}(e) \, \| \, R_{\text{pred}}(e) \, \| \, R_{\text{act}}(e)]$, where,

- $R_{\text{type}} \in \{0,1\}^2$: One-hot encoding of edge type (predicate-object or action-object)
- $R_{\text{pred}} \in \{0,1\}^m$: For predicate-object edges, one-hot encoding of argument position ($m$ is max predicate arity)
- $R_{\text{act}} \in \{0,1\}^{(m+|P|)}$: For action-object edges, concatenation of:
  - One-hot encoding of parameter position in action schema ($m$ bits)
  - Binary vector indicating which predicates are satisfied by the object in grounded action ($|P|$ bits). We only encode edge information for the actions that are applicable in the current state.

**Global Features:** The global node $g$ is initialized with a zero vector in $\mathbb{R}^h$ where $h$ is the chosen hidden representation dimension. This node can be used to aggregate and propagate graph-level information during message passing. This global node is essential in passing information at each round from nodes that are far apart. Hence, while nodes get their location information from neighboring edges, they also receive some global context at each round of the GNN ensuring that the number of GNN rounds does not become a bottleneck in passing information in large graphs. This is important since, as problems get larger the graph size grows, but the number of GNN rounds stays constant. The global node ensures all nodes and edges in the graph have some global context at each step.

This graph representation captures both the structural and semantic information necessary for learning planning heuristics while maintaining a bounded feature dimension independent of problem size. The node features encode type and semantic information, while the edge features capture the relationships between objects, predicates, and actions in the planning domain.

### 3.3 Neural Architecture

Our neural architecture processes this graph representation through multiple components designed to handle the challenges of processing variable-sized inputs, capturing long range dependencies between objects and actions, and making sequential decisions to select actions and their arguments as shown in algorithm 1. We detail each component below:

#### 3.3.1 GNN Encoder

Graph Neural Networks (GNNs) [20] are particularly well-suited for encoding states in planning problems as they can naturally process relational structures while being invariant to permutations and

**Procedure** GABAR($G = (\mathcal{V}, \mathcal{E})$)

1: **Initialize:** $\{\mathbf{v}_i^0\}_{i\in\mathcal{V}}, \{\mathbf{e}_{ij}^0\}_{(i,j)\in\mathcal{E}}, \mathbf{g}^0$

2: *// GNN Encoder*

3: **for** $l = 0$ **to** $L - 1$ **do**

4:   *// Edge Update*

5:   $\forall(i,j) \in \mathcal{E}: \mathbf{e}_{ij}^{l+1} = \phi_e([\mathbf{e}_{ij}^l; \mathbf{v}_i^l; \mathbf{v}_j^l; \mathbf{g}^l])$

6:   *// Node Update*

7:   $\forall i \in \mathcal{V}: \mathbf{v}_i^{l+1} = \phi_v([\mathbf{v}_i^l; \text{AGG}(\{\mathbf{e}_{ij}^{l+1} | j \in \mathcal{N}(i)\}); \mathbf{g}^l])$

8:   *// Global Update*

9:   $\mathbf{g}^{l+1} = \phi_g([\mathbf{g}^l; \text{AGG}(\{\mathbf{v}_i^{l+1} | i \in \mathcal{V}\}); \text{AGG}(\{\mathbf{e}_{ij}^{l+1} | (i,j) \in \mathcal{E}\})])$

10: **return** DECODER($\mathbf{g}^L, \{\mathbf{v}_i^L\}_{i\in\mathcal{V}}, A, O, k$)

**Procedure** DECODER($\mathbf{g}^L, \{\mathbf{v}_i^L\}_{i\in\mathcal{V}}, A, O, k$)

1: $h_1 = \text{GRU}(\mathbf{g}^L, \mathbf{0})$

2: *// Select action schema*

3: $\text{score}(a) = \text{MLP}(h_1 \odot \mathbf{v}_a^L)$ for each $a \in A$

4: $a^* = \arg\max_{a\in A} \text{softmax}(\text{score}(a))$

5: $h_2 = \text{GRU}(h_1, \mathbf{v}_{a^*}^L)$

6: *// Select parameters*

7: **for** $i = 1$ **to** required parameter count **do**

8:   $\text{score}(o) = \text{MLP}(h_{i+1} \odot \mathbf{v}_o^L)$ for each $o \in O$

9:   $o_i^* = \arg\max_{o\in O} \text{softmax}(\text{score}(o))$

10:   $h_{i+2} = \text{GRU}(h_{i+1}, \mathbf{v}_{o_i^*}^L)$

11: **return** $(a^*, o_1^*, o_2^*, \ldots, o_n^*)$

**Algorithm 1:** Graph Attention-Based Action Ranking (GABAR)

handling varying input sizes [29]. This allows them to learn patterns that generalize across different problem instances within the same domain, regardless of the number of objects involved. The key insight is that planning states are inherently relational - objects interact through predicates and actions - and GNNs can capture these relationships through message passing between nodes and edges. The GNN encoder (Algorithm 1, left side) transforms the input graph into learned embeddings.

**Initialization** (line 1): We initialize node features, edge features, and a global feature vector from the input graph. **Message Passing** (lines 3-9): For $L$ iterations, the encoder performs the following three update steps. 1) *Edge Update* (line 5): Each edge updates its representation by combining features from its connected elements. 2) *Node Update* (lines 7-8): Each node aggregates information from its neighboring edges along with global context. 3) *Global Update* (lines 9-10): The global vector is updated by aggregating information from all nodes and edges.

The aggregation function (AGG) employs attention mechanisms to weight different elements according to their relevance. This architecture makes several key design choices motivated by the planning domain. First, we explicitly model and update edge representations because edges in our graph capture crucial information about action applicability. This edge information helps guide the model toward selecting valid and effective actions. Second, we include a global node that can rapidly propagate information across the graph. This is important because without the global node, information would need to flow through many message-passing steps to reach distant parts of the graph. The global node acts as a shortcut, allowing the model to maintain a comprehensive view of the planning state even as the number of objects and relations grows. After $L$ rounds of message passing have been performed, the final global node embedding $g^{l+1}$ captures the relevant planning context needed for action selection.

### 3.3.2 GRU Decoder

The GRU-based decoder (Algorithm 1, right side) uses the encoded graph to select an action and its parameters. **Initialization** (line 1): The decoder initializes its hidden state using the global graph embedding and a zero vector. **Action Schema Selection** (lines 3-5): The decoder computes a score for each potential action schema (line 3). It then Selects the highest-scoring action (line 4) and updates the hidden state with the selected action's features (line 5).

**Parameter Selection** (lines 7-10): For each required parameter position, scores are computed for all objects using the current hidden state (line 8). The highest-scoring object is selected as the parameter (line 9). The hidden state is updated with the selected object's features (line 10). *Output* (line 11): The decoder returns the complete action grounding.

This autoregressive approach ensures that each parameter selection is conditioned on the graph structure, the selected action schema, and all previously selected parameters. By updating the GRU hidden state with each selection, the model captures dependencies between different components

of the action. We repeat this greedily until all action parameters have been extracted. Since greedy parameter selection is often too myopic, we employ beam search to explore multiple choices in parallel. For a beam width $k$, at each step, we maintain the $k$ highest-scoring partial sequences. The final output is a ranked list of $k$ action groundings $(a, o_1, \ldots, o_n)$ along with their accumulated scores. This process of GRU-based decoding helps generalize the framework to handle actions with any arity, as we can extract as many objects as necessary for the action selected in the first step of the decoding. Hence, at any given state, we can use the decoder to extract the ranking over the set of actions with different arity all at once. This aligns with our goal of learning to rank fully grounded actions rather than independently optimizing each argument (extended beam search version of the decoder that maintains multiple candidate action groundings in parallel is provided as algorithm 2 in the Appendix). While training optimizes for selecting the optimal action, the learned model provides a ranking of the top $k$ grounded actions during execution. The planner then selects the first legal action in the current state according to the ranking. It also uses the ranking to avoid cycles - if an action leads to a state previously visited, then the next best action is considered.

### 3.4    Data Generation and Training

For each planning domain, we generate training data by solving a set of small problem instances using an optimal planner. Each training example consists of a planning state $s$, goal specification $G$, and the first action $a^*$ from the optimal plan from $s$ to $G$. For states with multiple optimal actions, we randomly select one to avoid biasing the model.

The state-goal pairs are converted into our graph representation $G = (V, E, X, R)$ as described in the graph representation section. For each action $a^*$ in the training data, we create supervision signals in the form of:

- $y_a$: A one-hot vector over the action schema space indicating the correct action type
- $y_o = \{y_{o1}, ..., y_{ok}\}$: A sequence of $k$ one-hot vectors over the object space, where $k$ is the maximum number of parameters any action can take, indicating the correct objects for each parameter position

For action schema selection, the model needs to learn to assign the highest score to the correct action schema among all possible schemas. For each parameter position, the model needs to learn to assign the highest score to the correct object among all candidate objects. This is done using the following loss function.

**Loss Function.** Given a training instance $(G, y_a, y_o)$, GABAR computes action scores $s_a$ for all possible action schemas and object scores $s_{o_i}$ for each parameter position $i$. The total loss is computed as $L = L_{action} + L_{objects}$, where, $L_{objects}$ is the sum of cross-entropy losses between $\text{softmax}(s_{o_i})$ and $y_{o_i}$ for each parameter position $i$.

**Training Procedure.** For all domains, we train the model using the Adam optimizer with a learning rate of $0.0005$, 9 rounds of GNN, and batch size of 16, hidden dimensionality of 64. Training proceeds for a maximum of 500 epochs, and we select the model checkpoint that achieves the lowest loss on the validation set for evaluation. It takes between 1-2 hours to train a model for each domain on an RTX 3080.

## 4    Experiments

We evaluate GABAR's performance across a diverse set of classical planning domains. Our experiments aim to assess both the quality of learned policies and their ability to generalize to significantly larger problems than those in training. We selected eight standard planning domains that present different types of structural complexity and scaling dimensions. The domains are : BlocksWorld, Gripper, Miconic, Spanner, Logistics, Rovers, Grid and Visitall (More details about domains and number of training instances in appendix A.1).

We divide the test set for each domain into 3 separate subsets, easy, medium, and hard with increasing difficulty with problem sizes as defined in table 1 along with the training and validation dataset sizes. Each test subset has 100 problems. In contrast, the training dataset consists of problems smaller and simpler than the ones found in the easy subset.

Table 1: Distribution of problem sizes across train, validation, and test datasets. The ranges indicate the number of objects/variables defining each domain's problem complexity.* In Visitall and Grid, size refers to number of cells

| domain | Train | Val | Test | | | domain | Train | Val | Test | | |
| --- | --- | --- | --- | --- | --- | --- | --- | --- | --- | --- | --- |
| | | | easy | med | hard | | | | easy | med | hard |
| Blocks | $[6, 9]$ | $[10]$ | $[11, 20]$ | $[21, 30]$ | $[31, 40]$ | Rovers | $[3, 9]$ | $[10]$ | $[11, 30]$ | $[31, 50]$ | $[51, 70]$ |
| Gripper | $[5, 15]$ | $[17]$ | $[20, 40]$ | $[41, 60]$ | $[61, 100]$ | Visitall* | $[9, 36]$ | $[49]$ | $[50, 100]$ | $[101, 200]$ | $[201, 400]$ |
| Miconic | $[1, 9]$ | $[10]$ | $[20, 40]$ | $[41, 60]$ | $[61, 100]$ | Grid* | $[25, 49]$ | $[63]$ | $[64, 100]$ | $[100, 125]$ | $[125, 154]$ |
| Spanner | $[2, 9]$ | $[10]$ | $[11, 20]$ | $[21, 30]$ | $[31, 40]$ | Logistics | $[4, 7]$ | $[8]$ | $[15, 20]$ | $[21, 25]$ | $[26, 30]$ |

We evaluate GABAR against three state-of-the-art approaches for learning generalized policies: GPL, ASNets, and GRAPL. Each represents a different architectural paradigm for tackling generalized planning.

## 4.1 Baseline Approaches

**GPL** (Generalized Policy Learning) [24] learns value functions over states using GNNs. It selects actions by identifying unvisited states with the lowest estimated cost-to-goal. GPL attempts to learn a globally consistent value function by minimizing the regularized Bellman error $|V(s) - (1 + \min_{s' \in N(s)} V(s'))| + \max\{0, V^*(s) - V(s)\} + \max\{0, V(s) - \delta V^*(s)\}$ where $s$ is state and $V(s)$ is value function defined as cost to goal.

**ASNets** (Action Schema Networks) [27] employs a neural network with alternating action and proposition layers. Each layer contains modules that connect only to related modules in adjacent layers. ASNets shares weights across modules of the same action schema or predicate, enabling generalization across problems of varying sizes but limiting its reasoning to a fixed-depth receptive field.

**GRAPL** (Graph Relational Action Ranking for Policy Learning) [16] learns to rank actions using canonical abstractions where objects with identical properties are grouped together. Its neural network predicts both action probabilities and plan length estimates but lacks the explicit action-object graph structure and conditional decoding mechanisms of our approach.

**Ablations**: To assess each component's contribution to GABAR, we conducted ablation experiments with three variants:

- **GABAR-ACT**: Removes action nodes and action-object edges from the graph, using only object and predicate nodes for decision-making.
- **GABAR-CD**: Eliminates conditional decoding by selecting action parameters independently rather than sequentially. This tests the importance of dependencies between action parameters.
- **GABAR-RANK**: Replaces action ranking with a value function learning objective while maintaining our graph representation.

These ablations isolate our key innovations—action-centric graphs, GRU-based conditional decoding, and action ranking—to measure their individual impact on performance.

**Evaluation Metrics.** We evaluate GABAR using two primary metrics:

- **Coverage (C):** The percentage of test instances successfully solved within a 1000-step limit.
- **Plan Quality Ratio(P):** Ratio of plan length produced by Fast Downward (FD) [14] planner to the plan length produced by the learned policy. FD is run with `fd-lama-first` setting. We chose this satisficing configuration over optimal planners since optimal planners fail to solve most test problems within a reasonable time. While this means we cannot guarantee the optimality of the reference plans, it provides a practical baseline for assessing solution quality.

High coverage on larger instances demonstrates the model's ability to learn general action selection strategies, while plan quality ratio reveals whether these strategies produce efficient plans as problems scale up.

# 5 Results and Discussions

Table 2: Coverage (C) and plan quality ratio (P) across different domains and difficulty levels. Turqoise color intensity for C values indicates coverage score (Only values above 50% are highlighted: Light: 50-74%, Medium: 75-89%, Dark: 90-100%). P values are highlighted in violet with similar thresholds (No highlight: 0-0.49, Light: 0.5-0.74, Medium: 0.75-0.99, High: 1.0+). Columns are grouped into Baselines and Ablations. Combined rows show averages across all domains. (*) only non-zero values from their respective domains were considered

| | | Baselines | | | | | | GABAR | | Ablations | | | | | |
| | | GPL | | ASNets | | GRAPL | | | | GABAR-ACT | | GABAR-CD | | GABAR-RANK | |
| Domain | Diff | C↑ | P↑ | C↑ | P↑ | C↑ | P↑ | C↑ | P↑ | C↑ | P↑ | C↑ | P↑ | C↑ | P↑ |
|---|---|---|---|---|---|---|---|---|---|---|---|---|---|---|---|
| Blocks | E | **100** | 1.1 | **100** | **1.6** | 64 | 0.65 | **100** | 1.5 | 44 | 0.65 | **100** | 0.92 | 29 | 0.79 |
| | M | 45 | 0.68 | **100** | 1.5 | 48 | 0.44 | **100** | **1.6** | 14 | 0.49 | 92 | 0.81 | 21 | 0.71 |
| | H | 10 | 0.33 | 92 | 1.4 | 38 | 0.28 | **100** | **1.7** | 4 | 0.35 | 81 | 0.80 | 9 | 0.61 |
| Miconic | E | 97 | 0.97 | **100** | **1.0** | 68 | 0.56 | **100** | **1.0** | 35 | 0.55 | 97 | 0.88 | 42 | 0.67 |
| | M | 37 | 0.56 | **100** | **0.98** | 65 | 0.54 | **100** | 0.97 | 18 | 0.33 | 94 | 0.86 | 29 | 0.37 |
| | H | 19 | 0.29 | 90 | 0.92 | 60 | 0.49 | **100** | **0.95** | 2 | 0.27 | 88 | 0.83 | 16 | 0.29 |
| Spanner | E | 73 | **1.1** | 78 | 0.86 | 22 | 0.65 | **94** | **1.1** | 31 | 0.65 | 87 | 0.98 | 57 | 0.82 |
| | M | 42 | 0.56 | 60 | 0.69 | 5 | 0.55 | **93** | **0.99** | 11 | 0.27 | 81 | 0.93 | 42 | 0.77 |
| | H | 3 | 0.18 | 42 | 0.61 | 0 | - | **89** | **0.91** | 0 | - | 62 | 0.79 | 12 | 0.45 |
| Gripper | E | **100** | 1.0 | 78 | 0.98 | 26 | 0.95 | **100** | **1.1** | 31 | 0.56 | 95 | 1.0 | 55 | 0.58 |
| | M | 56 | 0.85 | 54 | 0.91 | 12 | 0.67 | **100** | **0.99** | 23 | 0.40 | 92 | 0.93 | 43 | 0.41 |
| | H | 21 | 0.74 | 42 | 0.88 | 0 | - | **100** | **0.96** | 9 | 0.28 | 87 | 0.86 | 21 | 0.33 |
| Visitall | E | 69 | **1.3** | **94** | 0.96 | 92 | 1.1 | 93 | 1.1 | 72 | 1.2 | 91 | 1.1 | 52 | 0.64 |
| | M | 15 | 0.76 | 86 | 0.93 | 88 | 1.0 | **91** | 1.0 | 64 | 0.93 | 89 | **1.1** | 46 | 0.56 |
| | H | 0 | 0 | 64 | 0.81 | 78 | 0.99 | **88** | 1.1 | 44 | 0.67 | 83 | **1.2** | 39 | 0.54 |
| Grid | E | 74 | 0.89 | 52 | 0.81 | 20 | 0.38 | **100** | **0.91** | 21 | 0.56 | 79 | 0.87 | 17 | 0.54 |
| | M | 17 | 0.61 | 45 | 0.66 | 3 | 0.28 | **97** | **0.85** | 8 | 0.46 | 71 | 0.65 | 12 | 0.28 |
| | H | 0 | 0 | 21 | 0.60 | 0 | - | **92** | **0.74** | 0 | - | 54 | 0.53 | 0 | - |
| Logistics | E | 56 | 0.61 | 39 | 0.71 | 32 | 0.81 | **90** | 0.75 | 12 | 0.64 | 31 | **0.86** | 41 | 0.65 |
| | M | 7 | 0.21 | 22 | 0.55 | 9 | 0.45 | **76** | **0.65** | 3 | 0.49 | 25 | 0.54 | 21 | 0.49 |
| | H | 0 | 0 | 4 | 0.39 | 0 | - | **71** | **0.59** | 0 | - | 6 | 0.35 | 0 | - |
| Rovers | E | 64 | 0.99 | 67 | 0.96 | 21 | 0.35 | **87** | **1.0** | 22 | 0.75 | 44 | 0.81 | 33 | 0.67 |
| | M | 9 | 0.32 | 56 | 0.87 | 5 | 0.19 | **82** | **0.96** | 6 | 0.66 | 37 | 0.63 | 9 | 0.56 |
| | H | 0 | 0 | 31 | 0.64 | 0 | - | **77** | **0.97** | 0 | - | 19 | 0.57 | 0 | - |
| Combined | E | 79.1 | 0.98 | 76 | 0.98 | 43.5 | 0.67 | **95.5** | **1.04** | 33.5 | 0.69 | 78 | 0.93 | 40.2 | 0.67 |
| | M | 28.5 | 0.56 | 65.4 | 0.88 | 29.3 | 0.51* | **92.2** | **1.01** | 18.4 | 0.50 | 72.7 | 0.80 | 27.8 | 0.51 |
| | H | 6.5 | 0.39* | 48.5 | 0.78 | 22.1 | 0.58* | **89.2** | **0.99** | 7.4 | 0.39* | 60 | 0.73 | 12.1 | 0.44* |

Table 2 shows the performance of GABAR, its ablations, and the baseline approaches across all domains. The results demonstrate significant differences in generalization capabilities and solution quality.

## 5.1 Generalization Performance

GABAR generalizes remarkably well across all domains, with coverage dropping minimally from 95.5% on easy problems to 89.5% on hard problems. This generalization is particularly evident in domains like Visitall, where GABAR solves problems 8 times larger than those in training. On BlocksWorld, Gripper, and Miconic, GABAR achieves 100% success rate at all difficulty levels.

Even in complex domains requiring multi-relation features (Grid, Logistics) or higher arity predicates (Rovers), GABAR maintains strong performance with success rates of 92%, 62%, and 77%

respectively on hard instances. This demonstrates that our graph structure effectively encodes critical problem information, and the decoder successfully captures relationships between actions and their parameters. We now analyze GABAR against baselines and relevant ablation results, highlighting how our design choices contribute to performance improvements.

**Action Ranking vs. Value Function Learning:** GPL uses GNNs to learn value functions over states, aiming to produce globally consistent estimates of cost-to-goal [24]. As problems grow larger, maintaining these consistent estimates becomes significantly more challenging. GPL's coverage drops to 0% on hard instances of Logistics and Visitall where GABAR maintains 71% and 88% coverage. Our ablation GABAR-RANK, which replaces action ranking with value function learning while keeping our graph structure intact, shows a similar pattern—coverage decreases from 89.2% to just 12.1% on hard problems. The PQR also drops from 0.98 in easy problems to 0.39 in hard problems for GPL, 0.67 to 0.44 for GABAR-RANK. This parallel between GPL and GABAR-RANK confirms that directly ranking actions is more effective than learning value functions.

**Conditional vs. Non-conditional Decoding:** GRAPL also learns action rankings but uses an abstract state representation without explicit modeling of action parameter dependencies [16]. While conceptually similar to our approach, GRAPL achieves only moderate performance on easy instances (43.5% coverage) and struggles on harder problems (22.1% coverage) compared to GABAR's 95.5% and 89.2%. Our ablation GABAR-CD, which removes our conditional decoding mechanism, shows a similar degradation—coverage drops from 89.2% to 60.0% on hard problems overall, with particularly dramatic decreases in complex domains like Logistics (from 62% to 21%). This comparison highlights the critical importance of our conditional GRU-based decoder for handling complex dependencies in domains where objects participate in multiple predicates simultaneously (e.g., a package being in both a city and a vehicle). The decoder learns relationships between parameter selections by conditioning each object choice on previously selected objects, enabling the model to capture complex inter-parameter relationships that GRAPL cannot.

**Action-Centric Graph Structure:** ASNets employs alternating action and proposition layers with a fixed-depth receptive field [27], limiting its ability to reason about long chains of dependencies. While ASNets performs reasonably on medium-difficulty problems, it generalizes poorly to larger instances—achieving only 21%, 4%, and 31% coverage on hard instances of Grid, Logistics, and Rovers, compared to GABAR's 92%, 62%, and 77%. Although we don't have a perfect ablation parallel for ASNets, our GABAR-ACT ablation (removing action nodes) shows catastrophic performance drops from 95.5% to 33.5% on easy problems and from 89.2% to just 7.4% on hard problems. This suggests that explicit action-object relationship modeling in the graph structure is critical for robust generalization.

These comparisons demonstrate that GABAR's superior performance stems from the synergistic combination of three key ideas: (1) an action-centric graph representation that captures relationships between actions and objects, (2) conditional decoding that models dependencies between action parameters, and (3) direct action ranking that avoids the challenges of learning globally consistent value functions. The ablation studies confirm that each component addresses these limitations present in current approaches to generalized planning.

# 6 Conclusion and Future Work

We presented **GABAR**, a novel graph-based architecture for learning generalized policies in classical planning through action ranking. Our key contributions include (1) an action-centric graph representation that explicitly captures action-object relationships, (2) a GNN architecture augmented with global nodes and GRUs for effective information propagation, and (3) a sequential decoder that learns to construct complete grounded actions. First, GABAR shows strong generalization capabilities, successfully solving problems significantly larger than those used in training. The system maintains high coverage (89%) even on the hardest test instances that are up to 8 times larger than training problems. This generalization is particularly significant in complex domains like logistics and grid that require coordinating multiple entities. Second, our extensive baseline and ablation studies validate that this superior performance stems from the synergistic combination of our core design choices. Our comparison with the value-based GPL baseline and our GABAR-RANK ablation confirms that directly ranking local actions is a more robust and generalizable strategy for large problems than learning a globally consistent value function . Furthermore, the significant performance gap between

GABAR and both the GRAPL baseline and our GABAR-CD (non-conditional) ablation highlights the critical importance of the conditional decoder for handling complex action-parameter dependencies . Finally, comparisons against ASNets and the failure of the GABAR-ACT (no action nodes) ablation show that explicitly modeling action-object relationships within the graph is essential for robust generalization. The primary challenge for the proposed method is the graph representation's growth rate, particularly for domains with high-arity actions or predicates. Future work could explore more compact representations while maintaining expressiveness and investigate ways of pruning irrelevant actions in the graph. Other future work could also include expanding the representation to handle uncertainty and solve more real-world problems such as object rearrangement.

## 7 Acknowledgments

The authors acknowledge the support of Army Research Office under grant W911NF2210251

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

# A Appendix

## A.1 Domain descriptions

**Blocks World** involves manipulating blocks to achieve specific tower configurations. The domain's complexity scales with the number of blocks (6-9 blocks for training/validation, 10-40 blocks for testing).

Table 3: Number of training datapoints per domain

| Domain | Training Datapoints |
| --- | --- |
| Blocksworld | 3,348 |
| Gripper | 6,287 |
| Miconic | 4,458 |
| Spanner | 3,801 |
| Logistics | 6,556 |
| Rovers | 3,874 |
| VisitAll | 3,654 |
| Grid | 6,880 |

**Gripper** requires a robot with two grippers to transport balls between rooms. The domain scales primarily with the number of balls to be moved (5-17 balls for training/validation, up to 100 balls for testing).

**Miconic** involves controlling an elevator to transport passengers between floors. The domain complexity increases along two dimensions: the number of passengers (1-10 for training/validation, 20-100 for testing) and the number of floors (2-20 for training/validation, 11-30 for testing).

**Logistics** involves transporting packages between locations using trucks (for intra-city transport) and airplanes (for inter-city transport). The domain scales with both the number of cities (4-8 for training/validation, 15-30 for testing) and packages (3-9 for training/validation, 10-24 for testing).

**Visitall** requires an agent to visit all cells in a grid. The domain scales with grid size (9-49 cells for training/validation, up to 400 cells for testing - testing problems 8 times larger than the training dataset).

**Grid** involves navigating through a grid where certain doors are locked and require specific keys to open. The number of locks (3) and keys (5) remain the same across training and testing while varying the size of the grid ($7 \times 9$ for training/validation to $11 \times 14$ for testing - a 150% increase in cells).

**Spanner** requires tightening nuts at one end of a corridor with spanners collected along the way. Number of locations varies from 10-20 in training and testing, the number of spanners varies from 2-10 in training and 11 to 40 in testing.

**Rovers** requires multiple rovers equipped with different capabilities to perform experiments and send results back to the lander. Training and validation instances use 2-5 rovers and 3-10 waypoints; those for testing have 3-7 rovers and 11-70 waypoints.

Table 3 contains information about number of datapoints used for each domain [1]. All problems were generated using openly available PDDL-generators [21].

## A.2 Ablations

We present more ablations on our method in this section to further analyze the contribution of different components in our GABAR architecture. The two configurations tested are:

- **GABAR-G**: An ablated version of GABAR that removes the global node from the graph neural network. Without the global node, the model lacks a centralized representation that aggregates information across the entire state, potentially limiting its ability to reason about complex relationships spanning multiple objects.
- **GABAR-ACT_CD**: A severely limited version that removes both the action representation from the graph and the conditional decoding mechanism. This baseline lacks the capability to explicitly represent actions in the state graph and cannot condition parameter selection on previously chosen actions, effectively testing the importance of both structured action representation and sequential decision-making. Results Discussion

---

[1]Code : https://github.com/Learning-for-Seq-Decision-Making/ GABAR-Graph-based-action-ranking-for-planning. Dataset submitted as supplementary material

Table 4: Performance across domains and difficulty levels. Coverage (C) indicates percentage of test instances solved within 1000 steps. Plan Quality Ratio (P) reported only for solved instances. Turqoise color intensity for C values indicates coverage score (Only values above 50% are highlighted). P values are highlighted in violet with similar thresholds. Combined rows show averages across all domains. (*) only non-zero values from their respective domains were considered.

| Domain | Diff | ACT_CD | | GABAR | | GABAR-G | |
|---|---|---|---|---|---|---|---|
| | | C↑ | P↑ | C↑ | P↑ | C↑ | P↑ |
| Blocks | E | 11 | 0.33 | **100** | 1.5 | **100** | 1.2 |
| | M | 8 | 0.31 | **100** | **1.6** | 88 | 1.2 |
| | H | 0 | - | **100** | **1.7** | 32 | 0.70 |
| Miconic | E | 8 | 0.32 | **100** | 1.0 | 95 | 0.95 |
| | M | 3 | 0.27 | **100** | 0.97 | 70 | 0.87 |
| | H | 0 | - | **100** | **0.95** | 56 | 0.73 |
| Spanner | E | 12 | 0.43 | **94** | **1.1** | 82 | 0.93 |
| | M | 1 | 0.24 | **93** | **0.99** | 74 | 0.81 |
| | H | 0 | - | **89** | **0.91** | 54 | 0.62 |
| Gripper | E | 15 | 0.58 | **100** | 1.1 | 95 | 1.0 |
| | M | 5 | 0.34 | **100** | **0.99** | 92 | 0.95 |
| | H | 0 | - | **100** | **0.96** | 77 | 0.90 |
| Visitall | E | 23 | 0.58 | **93** | **1.1** | 77 | 0.96 |
| | M | 19 | 0.55 | **91** | **1.0** | 66 | 0.83 |
| | H | 16 | 0.48 | **88** | **1.1** | 29 | 0.51 |
| Grid | E | 0 | - | **100** | **0.91** | 79 | 0.83 |
| | M | 0 | - | **97** | **0.85** | 68 | 0.62 |
| | H | 0 | - | **92** | **0.74** | 33 | 0.41 |
| Logistics | E | 0 | - | **90** | **0.75** | 45 | 0.43 |
| | M | 0 | - | **76** | **0.65** | 22 | 0.33 |
| | H | 0 | - | **71** | **0.59** | 4 | 0.21 |
| Rovers | E | 3 | 0.45 | **87** | **1.0** | 77 | 0.85 |
| | M | 0 | - | **82** | **0.96** | 72 | 0.79 |
| | H | 0 | - | **77** | **0.97** | 55 | 0.65 |
| Combined | E | 9 | 0.44* | **95.5** | **1.04** | 80.2 | 0.89 |
| | M | 4.5 | 0.34* | **92.2** | **1.01** | 69.2 | 0.79 |
| | H | 2 | 0.48* | **89.2** | **0.99** | 42.5 | 0.59 |

The results from Table 4 clearly demonstrate the effectiveness of our complete GABAR architecture compared to its ablated counterparts:

Overall Performance: GABAR significantly outperforms both ablated versions across all domains and difficulty levels. With average coverage of 95.5%, 92.2%, and 89.2% for easy, medium, and hard problems respectively, GABAR demonstrates robust generalization capabilities.

**GABAR v/s GABAR-G :** The dramatic performance drop from GABAR to GABAR-G (especially on hard problems where coverage falls from 89.2% to 42.5%) highlights the critical importance of the global node in our architecture. This component enables effective information aggregation across the entire state graph, substantially improving the model's ability to handle complex planning scenarios. This degradation in quality suggests that the global node plays a crucial role in helping the model learn strategic action selection rather than just locally reasonable choices.

**GABAR v/s GABAR-ACT_CD :** The extremely poor performance of GABAR-ACT_CD (with coverage below 10% in most domains and near-zero on harder problems) confirms that both action representation within the graph and conditional decoding are essential components. Without them, the

model fails to capture the relationship between actions and their parameters or to maintain consistency in sequential decisions.

Plan Quality: Beyond just solving more problems, GABAR consistently generates higher-quality plans (as measured by PQR) compared to its ablated counterparts. This indicates that the full model not only finds solutions but discovers more efficient paths to the goal.

## A.3 LLM Comparisons

To evaluate the relative capabilities of our approach against state-of-the-art language models, we conducted experiments using OpenAI's O3 model and Gemini-2.5-Pro (both released in 2025). Following the methodology of [28], we adopted their One-Shot prompting technique to generate prompts for our planning problems. We extracted the generated plans from model responses and validated them against problem constraints.

Table 5: Comparative performance of GABAR against state-of-the-art Large Language Models (OpenAI-O3 and Gemini-2.5-Pro) using a One-Shot prompting methodology. GABAR substantially outperforms both LLMs in **Coverage (C↑)** and **Plan Quality (P↑)** across all difficulties. The performance gap widens significantly on medium (M) and hard (H) problems, where LLM coverage collapses, highlighting their limitations in complex problems. Turqoise color intensity for C values indicates coverage score (Only values above 50% are highlighted). P values are highlighted in violet with similar thresholds. Combined rows show averages across all domains. (*) only non-zero values from their respective domains were considered.

| Domain | Diff | OpenAI-O3 | | Gemini2.5-Pro | | GABAR | |
|---|---|---|---|---|---|---|---|
| | | C↑ | P↑ | C↑ | P↑ | C↑ | P↑ |
| Blocks | E | 73 | 1.03 | 81 | 1.1 | **100** | **1.5** |
| | M | 41 | 0.95 | 47 | 0.86 | **100** | **1.6** |
| | H | 4 | 0.61 | 12 | 0.81 | **100** | **1.7** |
| Miconic | E | 56 | 0.81 | 79 | 0.86 | **100** | **1.0** |
| | M | 12 | 0.69 | 36 | 0.58 | **100** | **0.97** |
| | H | 0 | - | 12 | 0.51 | **100** | **0.95** |
| Spanner | E | 38 | 0.81 | 42 | 0.75 | **94** | **1.1** |
| | M | 13 | 0.77 | 10 | 0.64 | **93** | **0.99** |
| | H | 0 | - | 0 | - | **89** | **0.91** |
| Gripper | E | 39 | 0.89 | 55 | 0.95 | **100** | **1.1** |
| | M | 7 | 0.75 | 12 | 0.81 | **100** | **0.99** |
| | H | 0 | - | 0 | - | **100** | **0.96** |
| Visitall | E | 37 | 0.88 | 43 | 0.97 | **93** | **1.1** |
| | M | 18 | 0.73 | 27 | 0.96 | **91** | **1.0** |
| | H | 0 | - | 0 | - | **88** | **1.1** |
| Grid | E | 22 | 0.81 | 26 | 0.67 | **100** | **0.91** |
| | M | 7 | 0.71 | 13 | 0.56 | **97** | **0.85** |
| | H | 0 | - | 0 | - | **92** | **0.74** |
| Logistics | E | 5 | 0.77 | 13 | 0.68 | **90** | **0.75** |
| | M | 0 | - | 0 | - | **76** | **0.65** |
| | H | 0 | - | 0 | - | **71** | **0.59** |
| Rovers | E | 12 | 0.78 | 43 | 0.45 | **87** | **1.0** |
| | M | 0 | - | 13 | 0.33 | **82** | **0.96** |
| | H | 0 | - | 0 | - | **77** | **0.97** |
| Combined | E | 33.4 | 0.85 | 44.0 | 0.8 | **95.5** | **1.04** |
| | M | 11.6 | 0.77* | 17.1 | 0.68* | **92.2** | **1.01** |
| | H | 0.4 | 0.61* | 1.5 | 0.51* | **89.2** | **0.99** |

Table 5 presents the comparative results. GABAR demonstrates substantially superior performance across all difficulty levels. On easy problems, while Gemini-2.5-Pro achieves the highest coverage among LLMs at 44.0%, GABAR reaches 95.5% coverage with better plan quality (1.04 vs 0.80). OpenAI-O3 achieves 33.4% coverage with a plan quality of 0.85.

The performance gap widens substantially on medium-difficulty problems. Gemini-2.5-Pro maintains 17.1% coverage while OpenAI-O3 drops to 11.6%, compared to GABAR's 92.2%. Both LLMs also show degraded plan quality (0.68-0.77) while GABAR maintains 1.01. This suggests that as planning complexity increases, language models struggle with both solution generation and plan quality.

On hard problems, both language models demonstrate severe limitations. OpenAI-O3 achieves only 0.4% coverage (solving problems in a single domain: Blocks), while Gemini-2.5-Pro reaches 3.0% (succeeding in Blocks and Miconic). Their plan quality on these rare successes remains substantially lower (0.61 and 0.51 respectively) compared to GABAR's consistent 0.99 across all domains.

These results highlight fundamental differences in approach. While LLMs rely on pattern matching from training data and struggle with systematic reasoning over complex state spaces, GABAR's architecture explicitly learns structural relationships between actions, objects, and predicates through its graph-based representation. This structural inductive bias, combined with conditional decoding mechanisms, enables GABAR to maintain robust performance as problem complexity scales—a critical requirement for practical planning applications.

## A.4   Plan Length and Coverage Analysis

In addition to coverage metrics, we analyze the efficiency of generated plans through plan length measurements. Plan length represents the number of actions in the solution found by each method, with shorter plans indicating more efficient solutions. However, raw plan length comparisons are fundamentally misleading when methods achieve different coverage rates because methods that solve fewer problems inherently select for simpler instances within each difficulty category, artificially deflating their average plan lengths. To address this selection bias, we report the Plan Quality Ratio (PQR) as our primary efficiency metric. PQR is computed as the ratio of the learned policy's plan length to Fast Downward's plan length, but crucially, only for problems successfully solved by both methods. This metric eliminates the confounding factor of different methods tackling different problem subsets—a method that solves only 10% of the easiest problems might show excellent raw plan lengths, but its PQR reveals the true efficiency when compared fairly on the same problem instances. For this reason, PQR serves as the main plan quality metric throughout our evaluation, providing meaningful comparisons even when coverage rates differ substantially between methods.

### A.4.1   Understanding Plan Length in Context of Coverage

We present the Plan length (PL) results for all baselines and GABAR in Table 6. GABAR demonstrates high efficiency when coverage differences are properly contextualized. While GABAR does not always achieve the absolute lowest plan lengths, its performance becomes exceptional considering that it consistently solves significantly more problems than competitors across the entire spectrum of problem complexity.

The apparent discrepancies in plan length are largely explained by coverage differences between methods. When GPL achieves a lower plan length of 27 on Spanner-Easy compared to GABAR's 31, GPL only covers 73% of problems while GABAR covers 94%. Similarly, on Visitall-Easy, GPL achieves a plan length of 79 with only 69% coverage, while GABAR produces plans of length 103 but solves 93% of problems. The additional problems that GABAR solves likely represent more complex instances requiring longer plans, explaining the higher average plan length.

When examining cases with comparable coverage, GABAR's plan length performance becomes more impressive. On Blocks-Medium and Blocks-Hard, where GABAR maintains 100% coverage compared to ASNets' 100% and 92% respectively, GABAR achieves superior plan lengths of 76 and 125 compared to ASNets' 79 and 156. This demonstrates that when solving the same problem set, GABAR produces more efficient plans than the baselines.

The combined results further reinforce this analysis. GABAR's average plan lengths of 76, 108, and 147 represent solutions to 95.5%, 92.2%, and 89.2% of all problems respectively. In contrast, competitors with seemingly better plan lengths like GPL (69, 148, 265) only solve 79.1%, 28.5%,

and 6.5% of problems. The dramatic coverage drop as difficulty increases explains why baselines maintain apparently stable plan lengths—they increasingly solve only the simplest instances while GABAR continues to tackle the complete problem set, including the most complex cases that naturally require longer plans.

This selection bias is precisely why PQR provides a more meaningful comparison. By evaluating plan quality only on problems that both the learned policy and Fast Downward successfully solve, PQR eliminates the confounding factor of different methods tackling different problem subsets. Even when the baselines achieve lower average plan lengths across their limited coverage, their PQR values remain inferior to GABAR's, confirming that GABAR's comprehensive coverage does not compromise solution quality when compared fairly on equivalent problem sets.

Table 6: Comparative performance of GABAR against baseline methods (GPL, ASNets, GRAPL). Metrics include **Coverage (C↑)**, **Plan Quality Ratio (P↑)**, and absolute **Plan Length (PL↓)**. Turqoise color intensity for C values indicates coverage score (Only values above 50% are highlighted). P values are highlighted in violet with similar thresholds. Combined rows show averages across all domains. (*) only non-zero values from their respective domains were considered.

| Domain | Diff | GPL C↑ | GPL P↑ | GPL PL↓ | ASNets C↑ | ASNets P↑ | ASNets PL↓ | GRAPL C↑ | GRAPL P↑ | GRAPL PL↓ | GABAR C↑ | GABAR P↑ | GABAR PL↓ |
|---|---|---|---|---|---|---|---|---|---|---|---|---|---|
| Blocks | E | 100 | 1.1 | 55 | 100 | 1.6 | **38** | 64 | 0.65 | 79 | 100 | 1.5 | 41 |
| | M | 45 | 0.68 | 137 | 100 | 1.5 | 79 | 48 | 0.44 | 214 | 100 | 1.6 | **76** |
| | H | 10 | 0.33 | 423 | 92 | 1.4 | 156 | 38 | 0.28 | 585 | 100 | 1.7 | **125** |
| Miconic | E | 97 | 0.97 | 167 | 100 | 1.0 | 162 | 68 | 0.56 | 252 | 100 | 1.0 | **160** |
| | M | 37 | 0.56 | 235 | 100 | 0.98 | **180** | 65 | 0.54 | 280 | 100 | 0.97 | 181 |
| | H | 19 | 0.29 | 448 | 90 | 0.92 | 209 | 60 | 0.49 | 329 | 100 | 0.95 | **202** |
| Spanner | E | 73 | 1.1 | **27** | 78 | 0.86 | 36 | 22 | 0.65 | 35 | 94 | 1.1 | 31 |
| | M | 42 | 0.56 | 61 | 60 | 0.69 | 55 | 5 | 0.55 | 50 | 93 | 0.99 | **44** |
| | H | 3 | 0.18 | 208 | 42 | 0.61 | 79 | - | - | - | 89 | 0.91 | **67** |
| Gripper | E | 100 | 1.0 | 82 | 78 | 0.98 | 76 | 26 | 0.95 | **61** | 100 | 1.1 | 78 |
| | M | 56 | 0.85 | 128 | 54 | 0.91 | **118** | 12 | 0.67 | 128 | 100 | 0.99 | 133 |
| | H | 21 | 0.74 | **139** | 42 | 0.88 | 131 | - | - | - | 100 | 0.96 | 156 |
| Visitall | E | 69 | 1.3 | **79** | 94 | 0.96 | 121 | 92 | 1.1 | 106 | 93 | 1.1 | 103 |
| | M | 15 | 0.76 | 188 | 86 | 0.93 | 194 | 88 | 1.0 | 181 | 91 | 1.0 | **179** |
| | H | - | - | - | 64 | 0.81 | 333 | 78 | 0.99 | 272 | 88 | 1.1 | **243** |
| Grid | E | 74 | 0.89 | **41** | 52 | 0.81 | 47 | 20 | 0.38 | 77 | 100 | 0.91 | 45 |
| | M | 17 | 0.61 | **66** | 45 | 0.66 | 73 | 3 | 0.28 | 143 | 97 | 0.85 | 63 |
| | H | - | - | - | 21 | 0.60 | 101 | - | - | - | 92 | 0.74 | **98** |
| Logistics | E | 56 | 0.61 | 117 | 39 | 0.71 | 101 | 32 | 0.81 | **88** | 90 | 0.75 | 127 |
| | M | 7 | 0.21 | 305 | 22 | 0.55 | **138** | 9 | 0.45 | 169 | 76 | 0.65 | 159 |
| | H | - | - | - | 4 | 0.39 | **217** | - | - | - | 71 | 0.59 | 232 |
| Rovers | E | 64 | 0.99 | **17** | 67 | 0.96 | 19 | 21 | 0.35 | 44 | 87 | 1.0 | 21 |
| | M | 9 | 0.32 | 78 | 56 | 0.87 | 36 | 5 | 0.19 | 125 | 82 | 0.96 | **33** |
| | H | - | - | - | 31 | 0.64 | 55 | - | - | - | 77 | 0.97 | **45** |
| Combined | E | 79.1 | 0.98 | **69** | 76.0 | 0.98 | 73 | 43.5 | 0.67 | 91 | 95.5 | 1.04 | 76 |
| | M | 28.5 | 0.56 | 148 | 65.4 | 0.88 | **111** | 29.3 | 0.51 | 172 | 92.2 | 1.01 | 108 |
| | H | 6.5 | 0.39* | 265 | 48.5 | 0.78 | 158 | 22.1 | 0.58* | 276 | 89.2 | 0.99 | **147** |

## A.5 Beam Search Action Decoder

The beam search action decoder (Algorithm 2) enhances the basic greedy decoder by maintaining multiple candidate sequences at each decoding step.

**Initialization** (line 1): Similar to the greedy approach, the decoder initializes its hidden state using the global graph embedding and a zero vector.

**Action Schema Selection** (lines 2-5): Instead of selecting only the highest-scoring action schema, beam search maintains the top-$k$ action schemas. For each action schema $a \in A$, a score $s_a$ is computed (line 3). The $k$ highest-scoring actions form the initial beam $B_0$ (lines 4-5), each representing a partial sequence with an action schema, an empty parameter list, the initial hidden state, and the action score.

**Parameter Selection with Beam Search** (lines 6-15): During each decoding step, the algorithm expands all sequences in the current beam by considering the top-$k$ object candidates for the next parameter position:

- For each sequence in the current beam $B_{t-1}$ (line 9), scores are computed for all objects using the current hidden state (line 10).
- The top-$k$ objects $O^*$ are identified (line 11).
- Each sequence is expanded with each of these top-$k$ objects (lines 12-14), creating $k \times |B_{t-1}|$ new candidate sequences in $B'_t$.
- The hidden state is updated for each new sequence (line 13).
- The beam is pruned to keep only the $k$ highest-scoring expanded sequences (line 15), forming the new beam $B_t$.

**Output** (line 16): After processing all parameter positions, the algorithm returns the set of completed sequences, each forming a fully grounded action.

In our implementation, we set $k = 2$ for the beam search. This parameter choice provides a reasonable trade-off between exploration and computational efficiency. A beam width of 2 allows the decoder to maintain alternative sequences when faced with uncertainty, which is particularly important for handling invalid actions that might need to be discarded. Setting a larger beam width provides better results theoretically but increases computational cost without proportional benefits in our domains.

---

**Procedure** ACTIONDECODERBEAM($\mathbf{g}^L, \{\mathbf{v}_i^L\}_{i \in \mathcal{V}}, A, O, k$)

---

1: $h_0 = \text{GRU}(\mathbf{g}^L, \mathbf{0})$

2: *// Step 1: Action schema selection*

3: $s_a = \text{MLP}(h_0 \odot \mathbf{v}_a^L), \forall a \in A$

4: $A^* = \{a_1, a_2, \ldots, a_k\} = \underset{a \in A}{\text{top-}k}(s_a)$

5: $B_0 = \{(a_j, [], h_0, s_{a_j}) | a_j \in A^*\}$

6: *// Step 2: Parameter selection with beam search*

7: **for** $t = 1$ **to** max_params **do**

8:    $B'_t = \emptyset$

9:    **for** $(a, [o_1, \ldots, o_{t-1}], h_{t-1}, s) \in B_{t-1}$ **do**

10:      $s_o = \text{MLP}(h_{t-1} \odot \mathbf{v}_o^L), \forall o \in O$

11:      $O^* = \{o_1, o_2, \ldots, o_k\} = \underset{o \in O}{\text{top-}k}(s_o)$

12:      **for** $o_j \in O^*$ **do**

13:        $h_t = \text{GRU}(h_{t-1}, \mathbf{v}_{o_j}^L)$

14:        $B'_t = B'_t \cup \{(a, [o_1, \ldots, o_{t-1}, o_j], h_t, s + s_{o_j})\}$

15:    $B_t = \underset{b \in B'_t}{\text{top-}k}(\text{score}(b))$

16: **return** $\{(a, [o_1, o_2, \ldots, o_n]) | (a, [o_1, o_2, \ldots, o_n], h, s) \in B_{\text{max\_params}}\}$

**Algorithm 2:** Beam Search Action Decoder

