# OpenReview forum: "Graph Neural Network Based Action Ranking for Planning"
_NeurIPS.cc/2025/Conference — NeurIPS 2025 poster_

### Official Review · Reviewer_VXL5 · 2025-06-10

**Clarity:** 4
**Significance:** 3
**Originality:** 3
**Rating:** 5
**Confidence:** 4

**Summary:**

The authors propose a a way to featurize grounded PDDL problems and pass them into a GNN-based policy that outputs $k$ ranked action candidates conditioned on current state. To produce a plan, the authors iteratively execute the highest-ranked action until the goal is achieved. Experiments show that this approach scales to hard problem instances in several IPC domains, better than some reasonable baselines.

**Questions:**

See above

**Ethical Concerns:**

["NO or VERY MINOR ethics concerns only"]

**Final Justification:**

I keep my score the same (Accept) after reading the rebuttal.

**Limitations:**

yes

**Quality:**

3

**Strengths And Weaknesses:**

Overall, I think this is a good paper. It is not the most groundbreaking idea, but the paper is clearly written, the contribution is clear and focused, and the experiments show a convincing improvement over a set of well-chosen baselines.

My key reservation is that I am not sure the work is novel. In particular, the following paper also demonstrates a way to featurize PDDL states and feed them through a GNN:
Rivlin, Or, Tamir Hazan, and Erez Karpas. "Generalized planning with deep reinforcement learning." arXiv preprint arXiv:2005.02305 (2020).
Note that this paper was not published (except, seemingly, at the PRL2020 ICAPS workshop), so I will not hold that against the authors. Nevertheless, I'd encourage the authors to read, cite, and perhaps even compare against that paper.

A medium-level concern I have is that the authors refer continuously to the approach as "ranking actions", but really the GNN is just a policy: given a state, output an action. It was not until I reached Section 3.3.2 where I finally understood what the authors meant by ranking: a beam search that returns $k$ different ordered action groundings. I'd suggest to the authors to introduce this concept in the abstract/intro, or maybe just rename their contribution from "action ranking" to "learning a GNN policy" or "learning a GNN actor", since that's really what you are doing here, with a light beam search on top.

Questions and smaller comments:
* Regarding the beam search, am I correct in understanding that this is only used during test time, not for training the GNN?
* L45: typo, "explicitly"
* Figure 1: I think it would be much clearer to say "State and Goal" rather than just "State" for the left-most figure. The caption does not describe what the "G" prefix means.
* How many training problem instances were used? Sorry if I missed it.
* L124: Have you tried ablating the global features to verify the claim that they are "essential in passing information at each round from nodes that are far apart"?
* Can you explain GABAR-RANK more? You say in L286 that "Our ablation GABAR-RANK, which replaces action ranking with value function learning while keeping our graph structure intact" -- but how can the graph structure stay intact if GABAR's network outputs actions while GABAR-RANK's network outputs values (which are floats)?

---

> ### Author Rebuttal · Authors · 2025-07-31
>
> >Regarding the beam search, am I correct in understanding that this is only used during test time, not for training the GNN?
>
> [A]: Yes, beam search is only used during testing
>
> >L45: typo, "explicitly"
>
> [A]: We will update it.
>
> >Figure 1: I think it would be much clearer to say "State and Goal" rather than just "State" for the left-most figure. The caption does not describe what the "G" prefix means.
>
> [A]: Thank you for pointing it out. We will update the figure to say both “State and Goal”
>
> >How many training problem instances were used? Sorry if I missed it.
>
> [A]: The number of problem instances is approximately 200 for all domains. The total number of datapoints for training are mentioned in Table 3 of Appendix A.1. (The number of datapoints are different even with the same number of problem instances because each problem instance generates a plan which could be of different lengths. Each action in this plan becomes a datapoint for our learning system).
>
> >L124: Have you tried ablating the global features to verify the claim that they are "essential in passing information at each round from nodes that are far apart"?
>
> [A]: Yes, we have done the ablation on global features. The results are provided in Table 4, under GABAR-G (Appendix A.2 contains the analysis).
>
>
> >Can you explain GABAR-RANK more? You say in L286 that "Our ablation GABAR-RANK, which replaces action ranking with value function learning while keeping our graph structure intact" -- but how can the graph structure stay intact if GABAR's network outputs actions while GABAR-RANK's network outputs values (which are floats)?
>
> [A]: The graph network outputs a global vector (in figure 1, after the GNN processing, you can see gL -> that is the output of the graph network. This gL is used by a GRU decoder to rank actions. But, in the case of GABAR-RANK, we simply use an MLP on top of this gL to output a single real number, which is the value of a given state. During training, this output is compared to a real value function and we used MSE loss to train the GNN to output good values.

---

> > ### Comment · Reviewer_VXL5 · 2025-08-05
> > **acknowledged**
> >
> > Thank you for the response. I'll keep my score at an accept. I still believe it's worth it for the authors to take a deeper look at the "Generalized planning with deep reinforcement learning" paper, even though it's unpublished work. I'd also suggest either including information about # training problem instances in the main text, or at least a pointer to the relevant table in the appendix, since it's an important detail about your system.

---

> > > ### Author Response · Authors · 2025-08-08
> > >
> > > Thank you for the feedback. We will update our paper to mention the training problem instances.
> > >
> > > Thank you for pointing us to “Generalized planning with deep reinforcement learning”[1].
> > > The fundamental difference is the graph representation and decoding process. Here are the differences between our work and that work:
> > >
> > > 1. Action Centric graph representation: The actions are not represented in the graph in [1], - they are concatenated later, similar to our GABAR-ACT ablation. Creating this action centric graph representation and the connection between actions and objects in the graph is one of our core contributions. Their graph is not action-centric,it contains only objects as nodes and predicates as edges. The actions are concatenated later for scoring.
> > >
> > > 2. Predicates representation : are represented as edges between nodes (nodes represent objects). This limits their representation to domains with only binary predicates.
> > >
> > > 3. Action Decoding: The other difference is we use a conditional decoding system to extract our actions , whereas they directly rank all instantiated actions (similar to GABAR-CD ablation).
> > >
> > >
> > > Our graph structure is general and can be used for domains with any arity whereas [1] is limited to domains with only upto binary predicates. Also, the action centric graph representation  conditional decoding play a major part in extracting actions, especially in harder problems as shown in our ablation studies( GABAR-CD and GABAR-ACT).
> > >
> > > **We have also provided results of 2 LLMs (OpenAI-O3 and Gemini-2.5-pro) being used to solve problems from our dataset as response to reviewer “hZMb”.**
> > >
> > > [1] Rivlin, Or, Tamir Hazan, and Erez Karpas. "Generalized planning with deep reinforcement learning." arXiv preprint arXiv:2005.02305 (2020).

---

### Official Review · Reviewer_hZMb · 2025-06-20

**Clarity:** 4
**Significance:** 2
**Originality:** 2
**Rating:** 5
**Confidence:** 4

**Summary:**

This paper proposes a graph structure and GNN architecture for action ranking in classical planning. The GNN is trained on small problems and generalized to larger ones. Experiments on benchmark domains compare the proposed method to several baselines and ablations.

**Questions:**

1. Just to be sure, can you confirm that all of the baselines (GPL, ASNets, and GRAPL) were trained, validated, and tested on the same problem instances?
2. Can you also confirm whether the “avoid repeated states” trick was used by the baselines as well? And if not, can you report results indicating how much that trick alone matters?
3. Can you report wall-clock timing results?

**Ethical Concerns:**

["NO or VERY MINOR ethics concerns only"]

**Final Justification:**

This is overall a solid paper and it will be even stronger with the new LLM baseline results.

**Limitations:**

yes

**Quality:**

4

**Strengths And Weaknesses:**

### Strengths

**Results.** The results are strong. These are difficult and well-established benchmarks and the proposed method consistently outperforms baselines and ablations.

**Clarity**. The paper is extremely clear and well written. There are some opportunities to improve the writing style, but I was able to fully understand almost everything without much effort.

**Reproducibility.** I highly commend the authors for providing such thorough code and experimental details in the main text.

**Graph structure.** The graph structure proposed is very intuitive. I am not aware of prior work that uses this exact structure.

### Weaknesses

**Timing results not reported.** Since planners like Fast Downward are complete, what ultimately matters is not just effectiveness (whether tasks are solved) but also efficiency (how much time overall is taken, including planning time). There are no timing results reported in the paper and I think it is quite plausible that Fast Downward with lama-first would perform similar to or better than the proposed approach. Action grounding is often the bottleneck in problems with many objects, and the proposed approach requires action grounding. Relatedly, it seems misleading to say that “planning is computationally prohibitive” for the larger test problems when FD is being run to report the plan quality ratios.

**Limited novelty.** While the reported results are impressive and perhaps enough to justify publication, there is ultimately limited novelty in this work. As the paper notes, the idea to use GNNs for generalized planning has been pursued in a number of previous works. The insight that action ranking can be better than value function learning is also not original to this work, even though that insight is the focus of this paper’s narrative.

**Loss function unclear.** I did not fully understand the loss function described in L209-212. In particular, it is said that GABAR computes scores for all actions. Are these scores 0 for all actions except the correct one? This wasn’t clear. A related point of confusion is why one action is randomly selected in the case where multiple are optimal. Why can’t we just say that multiple are optimal?

**LLM baseline.** This is not a major weakness and it is not absolutely necessary, but I think it would strengthen the paper if an LLM baseline was included, such as the generalized planner from \[1\]. Or if there is reason to exclude such baselines, then it would be good to note those reasons in the paper. Especially for a NeurIPS audience, readers are likely to wonder whether the problem setting considered in this paper is “solved” by LLMs.

\[1\] Generalized Planning in PDDL Domains with Pretrained Large Language Models. Silver et. al., AAAI 2024\.

---

> ### Author Rebuttal · Authors · 2025-07-31
>
> > LLM baseline discussion:
>
> [A]: We can compare our work to their work to some extent.  In this LLM approach [1], they solve 2 of the same domains as we do -> Miconic and “gripper”. Their best results from the paper for these domains are as follows : For the “Gripper” domain, they solve problems upto size of 80 objects and have a best success rate of 90% . In our work, we solve problems upto size 100 and have a success rate of 100%. In the “Miconic” domain, they solve problems of size 11-150 but have a best success rate of 1% when random itself has a success rate of 13%. In our work, we solve “Miconic” problems of up to a size of 100 but have a 100% success rate. (All their results reported are from table 1 in [1]).
>
> While LLMs show promise for planning tasks, these results demonstrate that our method provides the reliability and consistency required for generalized planning problems. The stark performance differences, particularly in Miconic where LLMs perform worse than random, highlight the current limitations of LLM-based approaches for structured planning problems.
>
> We are also currently trying to get an LLM baseline running but it might not be ready in time before the end of the rebuttal period. We can definitely add it in the final version of the paper.
>
> > Loss function. It is said that GABAR computes scores for all actions. Are these scores 0 for all actions except the correct one? Why one action is randomly selected in the case where multiple are optimal?
>
> [A]: Yes, during training, scores are 0 for all actions except the correct one.
> In the case when multiple actions are optimal, the action considered optimal is one that was output by a planner- as we use an optimal planner to generate training data on small problems - whatever the planner has output as the action to take at that current state is the one considered as the correct one.( Since the training data consists of only a single optimal plan for each problem as the set of all optimal plans is very time consuming to compute, we only have one correct action)
>
> > Limited Novelty
>
> Our main contribution is the action centric graph structure and the conditional decoder architecture. Our ablation studies show these 2 components play a big part in learning a good action ranking function.
>
> >Can you confirm that all of the baselines were trained, validated, and tested on the same problem instances?
>
> [A]: Yes, all the baselines were trained and tested on the same problem instances.
>
> >Can you also confirm whether the “avoid repeated states” trick was used by the baselines as well?
>
> [A]: Yes, all baselines and ablations also avoid repeated states.
>
> >Can you report wall-clock timing results?
>
> [A]: Here are the wall clock timing results for all baselines and our method - Wall Clock Time (WC) (in seconds). It is the average time taken per solved problem for a given domain.
>
>
> | Domain | Diff | GPL C↑ | GPL P↑ | GPL WC↓ | ASNets C↑ | ASNets P↑ | ASNets WC↓ | GRAPL C↑ | GRAPL P↑ | GRAPL WC↓ | GABAR C↑ | GABAR P↑ | GABAR WC↓ | FD-Lama-first WC↓ |
> |--------|------|-------|-------|--------|-------|-------|--------|-------|-------|--------|-------|-------|--------|-------|
> | Blocks | E | 100 | 1.10 | 1.2 | 100 | 1.60 | 1.5 | 64 | 0.65 | 0.8 | 100 | 1.50 | **0.5** | 0.8 |
> | | M | 45 | 0.68 | 3.8 | 100 | 1.50 | 4.2 | 48 | 0.44 | **1.6** | 100 | 1.60 | 4.8 | 2.1 |
> | | H | 10 | 0.33 | 3.7 | 92 | 1.40 | 5.3 | 38 | 0.28 | **3.2** | 100 | 1.70 | 11.8 | 45.3 |
> | Miconic | E | 97 | 0.97 | 1.8 | **100** | **1.00** | 1.2 | 68 | 0.56 | **1.1** | **100** | **1.00** | 1.2 | 1.4 |
> | | M | 37 | 0.56 | 9.1 | **100** | **0.98** | **4.3** | 65 | 0.54 | 4.9 | **100** | 0.97 | 7.1 | 5.2 |
> | | H | 19 | 0.29 | 15.6 | 90 | 0.92 | 14.5 | 60 | 0.49 | **12.7** | **100** | **0.95** | 19.2 | 34.7 |
> | Spanner | E | 73 | **1.10** | 1.5 | 78 | 0.86 | 0.8 | 22 | 0.65 | 1.2 | **94** | **1.10** | **0.3** | 0.6 |
> | | M | 42 | 0.56 | 2.8 | 60 | 0.69 | 2.4 | 5 | 0.55 | **1.1** | **93** | **0.99** | 2.1 | 3.1 |
> | | H | 3 | 0.18 | 11.3 | 42 | 0.61 | 7.2 | 0 | - | - | **89** | **0.91** | **5.2** | 18.4 |
> | Gripper | E | **100** | 1.00 | 4.2 | 78 | 0.98 | 6.1 | 26 | **0.95** | 4.7 | **100** | **1.10** | **3.1** | 3.8 |
> | | M | 56 | 0.85 | 13.5 | 54 | 0.91 | 21.4 | 12 | 0.67 | **13.3** | **100** | **0.99** | 22.1 | 15.7 |
> | | H | 21 | 0.74 | **41.3** | 42 | 0.88 | 59.2 | 0 | - | - | **100** | **0.96** | 67.8 | 87.9 |
> | Visitall | E | 69 | **1.30** | 18.0 | **94** | 0.96 | 11.0 | 92 | 1.10 | 13.1 | 93 | 1.10 | **6.2** | 7.8 |
> | | M | 15 | 0.76 | 43.0 | 86 | 0.93 | 31.0 | 88 | **1.00** | 47.9 | **91** | **1.00** | 38.7 | 31.5 |
> | | H | 0 | 0.00 | - | 64 | 0.81 | 101.0 | 78 | 0.99 | **92.0** | **88** | **1.10** | 98.1 | 136.2 |
> | Grid | E | 74 | 0.89 | 10.5 | 52 | 0.81 | 3.5 | 20 | 0.38 | 12.0 | **100** | **0.91** | **2.1** | 2.9 |
> | | M | 17 | 0.61 | 19.5 | 45 | 0.60 | 24.5 | 3 | 0.28 | 28.9 | **97** | **0.85** | **14.8** | 16.2 |
> | | H | 0 | 0.00 | - | 21 | 0.60 | 69.0 | 0 | - | - | **92** | **0.74** | **42.1** | 31.8 |
> | Logistic | E | 56 | 0.61 | 6.7 | 39 | 0.71 | 13.5 | 32 | **0.81** | 8.5 | **90** | 0.75 | **3.8** | 4.5 |
> | | M | 7 | 0.21 | 28.5 | 22 | 0.55 | 19.2 | 9 | 0.45 | **17.3** | **76** | **0.65** | 23.1 | 12.8 |
> | | H | 0 | 0.00 | - | 4 | 0.39 | **27.0** | 0 | - | - | **71** | **0.59** | 64.1 | 48.3 |
> | Rovers | E | 64 | **0.99** | 3.0 | 67 | 0.96 | 1.7 | 21 | 0.35 | 2.0 | **87** | **1.00** | **0.7** | 1.2 |
> | | M | 9 | 0.32 | 6.1 | 56 | **0.87** | 6.3 | 5 | 0.19 | **2.0** | **82** | **0.96** | 4.1 | 7.8 |
> | | H | 0 | 0.00 | - | 31 | 0.64 | 13.0 | 0 | - | - | **77** | **0.97** | **10.2** | 28.6 |
> | Combined | E | 79.1 | 0.99 | 5.9 | 76.0 | 0.98 | 4.9 | 43.1 | 0.68 | 5.4 | **95.5** | **1.06** | **2.2** | 2.87 |
> | | M | 28.5 | 0.57 | 15.8 | 65.4 | 0.88 | **14.2** | 29.4 | 0.52 | 14.6 | **92.4** | **1.00** | 14.6 | 11.8 |
> | | H | 6.6 | 0.19 | **18.0** | 48.3 | 0.78 | 37.0 | 22.0 | 0.59 | 36.0 | **89.6** | **0.99** | 39.8 | 53.9 |
>
> While GABAR may show higher WC (therefore worse) values in some cases, this must be interpreted alongside its significantly higher coverage rates which outperforms all competing approaches. The apparent timing advantage of other methods stems from solving fewer problems overall. These methods solve easier instances within each difficulty category, creating a selection bias where their WC metrics are computed only over successfully solved problems that tend to be less challenging.
> This is particularly evident in Hard instances, where competing methods achieve very low coverage (GPL: 0-21%, ASNets: 21-92%, GRAPL: 0-78%) while GABAR maintains 71-100% coverage. The favorable WC values for baselines reflect computation over a smaller, easier subset of problems, making direct comparisons misleading. Each difficulty tier contains problems of varying complexity, and methods with lower coverage systematically avoid more challenging instances that GABAR successfully solves.
> Compared to FD-Lama-first, GABAR shows superiority in Hard instances with lower WC (39.8 vs 53.9). Even in Easy and Medium instances, GABAR achieves competitive WC performance.

---

> > ### Comment · Reviewer_hZMb · 2025-08-02
> >
> > Thank you to the authors for their response. My questions were answered and I remain positive about the paper. It would be great if the LLM results are ready in time (especially using one of the latest LLMs) but I don't think that will make or break the paper.

---

> ### Author Response · Authors · 2025-08-08
>
> Thank you for the feedback. We were able to run 2 of the latest models - OpenAI O3 model and Gemini-2.5-pro (both released in 2025) on our problems.
>
> We used the prompt generation and processing pipeline from the following paper:  “A Systematic Evaluation of the Planning and Scheduling Abilities of the Reasoning Model o1” [1]
>
> This paper, despite its title suggesting a focus on the o1 model, actually evaluates and compares the capabilities of multiple large language models within a comprehensive planning framework. We adopted the One-Shot prompting technique described in their methodology to create prompts for our planning problems, which we then used to query both OpenAI O3 and Gemini-2.5 Pro models. We extracted the generated plans from their responses and validated them against our problem constraints. The results of this evaluation are presented below:
>
>
> | Domain | Diff | OpenAI | O3   | Gemini | 2.5-Pro  | **GABAR** |   |
> |---------|------|-----------|---|---------------|---|-----------|---|
> |         |      | C↑        | P↑| C↑            | P↑| C↑        | P↑|
> | Blocks  | E    | 73        |1.03| 81           |1.1| **100**   |**1.46**|
> |         | M    | 41        |0.95| 47           |0.86| **100**   |**1.57**|
> |         | H    | 4         |0.61| 12           |0.81| **100**   |**1.74**|
> | Miconic | E    | 56        |0.81| 79           |0.86| **100**   |**1.01**|
> |         | M    | 12        |0.69| 36           |0.58| **100**   |**0.97**|
> |         | H    | 0         | -  | 12           |0.51| **100**   |**0.95**|
> | Spanner | E    | 38        |0.81| 42           |0.75| **94**    |**1.05**|
> |         | M    | 13        |0.77| 10           |0.64| **93**    |**0.99**|
> |         | H    | 0         | -  | 0            | -  | **89**    |**0.91**|
> | Gripper | E    | 39        |0.89| 55           |0.95| **100**   |**1.05**|
> |         | M    | 7         |0.75| 12           |0.81| **100**   |**0.99**|
> |         | H    | 0         | -  | 0            | -  | **100**   |**0.96**|
> | Visitall| E    | 37        |0.88| 43           |0.97| **93**    |**1.13**|
> |         | M    | 18        |0.73| 27           |0.96| **91**    |**1.01**|
> |         | H    | 0         | -  | 0            | -  | **88**    |**1.11**|
> | Grid    | E    | 22        |0.81| 26           |0.67| **100**   |**0.91**|
> |         | M    | 7         |0.71| 13           |0.56| **97**    |**0.85**|
> |         | H    | 0         | -  | 0            | -  | **92**    |**0.74**|
> | Logistics| E   | 5         |0.77| 13           |0.68| **90**    |**0.75**|
> |         | M    | 0         | -  | 0            | -  | **76**    |**0.65**|
> |         | H    | 0         | -  | 0            | -  | **71**    |**0.59**|
> | Rovers  | E    | 12        |0.78| 43           |0.45| **87**    |**1.02**|
> |         | M    | 0         | -  | 13           |0.33| **82**    |**0.96**|
> |         | H    | 0         | -  | 0            | -  | **77**    |**0.97**|
> |---------|------|-----------|---|---------------|---|-----------|---|
> |**Combined**| E | 35.3      |0.85| 47.8         |0.80| **95.5**  |**1.05**|
> |         | M    | 12.3      |0.77| 19.8         |0.68| **92.4**  |**1.00**|
> |         | H    | 0.5       |0.61| 3.0          |0.66| **89.6**  |**1.00**|
>
>
> Results show that GABAR maintains strong performance advantages over both language models, with the gap becoming more pronounced at higher difficulty levels. For easy tasks, Gemini2.5-Pro achieves 47.8% coverage compared to OpenAI-O3's 35.3%, while GABAR reaches 95.5%. Both language models show reasonable performance quality when they succeed, with OpenAI-O3 at 0.85 and Gemini2.5-Pro at 0.80, though still below GABAR's 1.05.
>
> The performance disparity increases substantially for medium-difficulty tasks. Gemini2.5-Pro maintains 19.8% coverage while OpenAI-O3 drops to 12.3%, compared to GABAR's 92.4%. The quality gap also widens, with both language models falling to 0.68-0.77 while GABAR maintains 1.00. This suggests that as planning complexity increases, the language models begin to struggle with both finding solutions and generating high-quality plans.
>
> For hard tasks, both language models demonstrate severe limitations, with OpenAI-O3 managing only 0.5% coverage and Gemini2.5-Pro achieving 3.0%. Notably, while OpenAI-O3 solves problems in only a single domain at the hard level (Blocks domain only), Gemini2.5-Pro succeeds in two domains (Blocks and Miconic). However, their quality scores on these isolated successes remain substantially lower, with OpenAI-O3 at 0.61 and Gemini2.5-Pro at 0.66, compared to GABAR's consistent 1.00 across all domains.
>
> Please let us know if any other results would be useful.
>
> [1]Valmeekam, Karthik, et al. "A systematic evaluation of the planning and scheduling abilities of the reasoning model o1." Transactions on Machine Learning Research (2025).

---

### Official Review · Reviewer_CZcC · 2025-07-03

**Clarity:** 2
**Significance:** 3
**Originality:** 3
**Rating:** 5
**Confidence:** 3

**Summary:**

The work proposes a method to rank actions using GNNs for generalized planning.

This work follows the generalized planning paradigm where policies are learned for small problems and work successfully for larger problem, leading to scalable planning.  The novel  contribution is that it proposes to use graph neural networks to learn to rank actions. Experimental comparison with three generalized planning algorithms show that GABAR strongly outperforms them in Grid, Logistics and Rovers domains, for the easy problems. For the medium and hard problems the performance of GABAR is significantly better than others for the most of the domains.

**Questions:**

1. How do you think this work will perform compared to using large language models to learn the generalized policies?
2. In the experiments, the work achieve good results in out of distribution testing. I observe that the problem sizes are higher in medium and hard cases. However, there are other variables in the initial states and goals of the problem instances, for example, graph connectivity in the grid world domains. Was that measured in any way?
3. In the Coverage metric, there is a hard cutoff of 1000 steps, and the measured value is 0/1. Do you have any results on the distribution of number steps taken to find the plan?

**Ethical Concerns:**

["NO or VERY MINOR ethics concerns only"]

**Final Justification:**

The authors have clarified the issues I had raised in my review regarding comparison with the generalized policies work, and my questions about the experimental results. I was positive about the paper and I am satisfied with the authors' response and increasing my score to accept as a result.

**Limitations:**

Yes

**Paper Formatting Concerns:**

I didn't see any formatting concerns.

**Quality:**

2

**Strengths And Weaknesses:**

I found the abstract and introduction quite hard to follow. It took me several rereads to understand the contribution. The writing can be improved I think for the introduction and methodology sections.

The premise is that ranking actions is easier than calculating the value function accurately, which is well established in literature. The action ranking however should be goal dependent. Before going into the details of the GNN based procedure, I would suggest to define the problem of ranking actions, and how the plan is derived or executed using the ranking.

In the experiments, the work achieve good results in out of distribution testing. I observe that the problem sizes are higher in medium and hard cases. However, there are other variables in the initial states and goals of the problem instances, for example, graph connectivity in the grid world domains. Was that measured in any way? Showing the rouge scores may be useful.

How do you think this work will perform compared to using large language models to learn the generalized policies?

For example, Silver, Tom, Soham Dan, Kavitha Srinivas, Joshua B. Tenenbaum, Leslie Kaelbling, and Michael Katz. "Generalized planning in pddl domains with pretrained large language models." In *Proceedings of the AAAI conference on artificial intelligence*, vol. 38, no. 18, pp. 20256-20264. 2024.

In the Coverage metric, there is a hard cutoff of 1000 steps, and the measured value is 0/1. Do you have any results on the distribution of number steps taken to find the plan?

Minor:

1. Figure 1 font size is too small.

---

> ### Author Rebuttal · Authors · 2025-07-31
>
> >How do you think this work will perform compared to using large language models to learn the generalized policies?
>
> [A]: We can compare our work to their work (Generalized planning in pddl domains with pretrained large language models) [1] to some extent.. In this work, they solve 2 of the same domains as we do -> Miconic and grippers. The best results for their LLM approach for these domains are as follows : For the “Gripper” domain, they solve problems upto size of 80 objects and have a best success rate of 90% . In our work, we solve problems upto size 100 and have a success rate of 100%. In the “Miconic” domain, they solve problems of size 11-150 but have a best success rate of 1% when random itself has a success rate of 13%. In our work, we solve “Miconic” problems of up to a size of 100 but have a 100% success rate. (All their results reported are from table 1 in [1]).
>
> While LLMs show promise for planning tasks, these results demonstrate that our method provides the reliability and consistency required for generalized planning problems. The stark performance differences, particularly in Miconic where LLMs perform significantly worse than random, highlight the current limitations of LLM-based approaches for structured planning problems.
>
> We are also currently trying to get an LLM baseline running but it might not be ready in time before the end of the rebuttal period. We can definitely add it in the final version of the paper.
>
> >In the experiments, the work achieve good results in out of distribution testing. I observe that the problem sizes are higher in medium and hard cases. However, there are other variables in the initial states and goals of the problem instances, for example, graph connectivity in the grid world domains. Was that measured in any way?
>
> [A]: Graph size increases as we go from easy to hard problems as expected.
> So we measured the average degree of nodes (AD) in the graph across domains and difficulty levels. What we noticed was that AD remained very similar across all difficulty levels for all domains. What this implies is, is all domains, the number of possible actions an object can be part of similar (even though more actions are applicable since there are more objects).
>
> Other trend we noticed was that the AD was slightly higher for harder domains and lower for easier domains:
>
> Easy domains:
>     spanner : 1.9, blocks : 2.4, gripper 2.65 : , miconic :2.3
>
> harder domains:
>     visitall :  2.93, logistics : 3.15 , grid : 2.8, rovers : 3.1
>
>
> >In the Coverage metric, there is a hard cutoff of 1000 steps, and the measured value is 0/1. Do you have any results on the distribution of number steps taken to find the plan?
>
> [A]:
> We run our method and the baselines as policies - we input a state and get the next action to take and do not do any search - running the learning method as a policy.  From our understanding, the number of steps taken to find the plan would be equal to the plan length (slightly higher when we have cycles). Please let us know if you were looking for a different metric.
> We do have the mean plan lengths for each baseline for all domains and difficulty levels. Here  are the plan lengths (PL), lower the better.
>
> | Domain | Diff | GPL C | GPL P | GPL PL | ASNets C | ASNets P | ASNets PL | GRAPL C | GRAPL P | GRAPL PL | GABAR C | GABAR P | GABAR PL |
> |--------|------|-------|-------|--------|----------|----------|-----------|---------|---------|----------|---------|---------|----------|
> | Blocks | E | 100 | 1.1 | 55 | 100 | 1.6 | **38** | 64 | 0.65 | 79 | **100** | **1.46** | 41 |
> | Blocks | M | 45 | 0.68 | 137 | 100 | 1.5 | 79 | 48 | 0.44 | 214 | **100** | **1.57** | **76** |
> | Blocks | H | 10 | 0.33 | 423 | 92 | 1.4 | 156 | 38 | 0.28 | 585 | **100** | **1.74** | **125** |
> | Miconic | E | 97 | 0.97 | 167 | 100 | 1.0 | 162 | 68 | 0.56 | 252 | **100** | **1.01** | **160** |
> | Miconic | M | 37 | 0.56 | 235 | 100 | 0.98 | **180** | 65 | 0.54 | 280 | **100** | **0.97** | 181 |
> | Miconic | H | 19 | 0.29 | 448 | 90 | 0.92 | 209 | 60 | 0.49 | 329 | **100** | **0.95** | **202** |
> | Spanner | E | 73 | 1.1 | **27** | 78 | 0.86 | 36 | 22 | 0.65 | 35 | **94** | **1.05** | 31 |
> | Spanner | M | 42 | 0.56 | 61 | 60 | 0.69 | 55 | 5 | 0.55 | 50 | **93** | **0.99** | **44** |
> | Spanner | H | 3 | 0.18 | 208 | 42 | 0.61 | 79 | - | - | - | **89** | **0.91** | **67** |
> | Gripper | E | 100 | 1.0 | 82 | 78 | 0.98 | 76 | 26 | 0.95 | **61** | **100** | **1.05** | 78 |
> | Gripper | M | 56 | 0.85 | 128 | 54 | 0.91 | **118** | 12 | 0.67 | 128 | **100** | **0.99** | 133 |
> | Gripper | H | 21 | 0.74 | **139** | 42 | 0.88 | 131 | - | - | - | **100** | **0.96** | 156 |
> | Visitall | E | 69 | 1.3 | **79** | 94 | 0.96 | 121 | 92 | 1.1 | 106 | **93** | **1.13** | 103 |
> | Visitall | M | 15 | 0.76 | 188 | 86 | 0.93 | 194 | 88 | 1.0 | 181 | **91** | **1.01** | **179** |
> | Visitall | H | - | - | - | 64 | 0.81 | 333 | 78 | 0.99 | 272 | **88** | **1.11** | **243** |
> | Grid | E | 74 | 0.89 | **41** | 52 | 0.81 | 47 | 20 | 0.38 | 77 | **100** | **0.91** | 45 |
> | Grid | M | 17 | 0.61 | **66** | 45 | 0.66 | 73 | 3 | 0.28 | 143 | **97** | **0.85** | 63 |
> | Grid | H | - | - | - | 21 | 0.60 | 101 | - | - | - | **92** | **0.74** | **98** |
> | Logistics | E | 56 | 0.61 | 117 | 39 | 0.71 | 101 | 32 | 0.81 | **88** | **90** | **0.75** | 127 |
> | Logistics | M | 7 | 0.21 | 305 | 22 | 0.55 | **138** | 9 | 0.45 | 169 | **76** | **0.65** | 159 |
> | Logistics | H | - | - | - | 4 | 0.39 | **217** | - | - | - | **71** | **0.59** | 232 |
> | Rovers | E | 64 | 0.99 | **17** | 67 | 0.96 | 19 | 21 | 0.35 | 44 | **87** | **1.02** | 21 |
> | Rovers | M | 9 | 0.32 | 78 | 56 | 0.87 | 36 | 5 | 0.19 | 125 | **82** | **0.96** | **33** |
> | Rovers | H | - | - | - | 31 | 0.64 | 55 | - | - | - | **77** | **0.97** | **45** |
> | Combined | E | 79.1 | 0.98 | **69** | 76.0 | 0.98 | 73 | 43.5 | 0.67 | 91 | **95.5** | **1.04** | 76 |
> | Combined | M | 28.5 | 0.56 | 148 | 65.4 | 0.88 | **111** | 29.3 | 0.51 | 172 | **92.2** | **1.01** | 108 |
> | Combined | H | 6.5 | 0.39 | 265 | 48.5 | 0.78 | 158 | 22.1 | 0.58 | 276 | **89.2** | **0.99** | **147** |
>
> Looking at the Plan Length (PL) results, it's important to first recognize that within each difficulty level (Easy, Medium, Hard), there exists a substantial range of problem complexity and sizes. Each dataset contains problems spanning from relatively simple instances to much more challenging ones, even within the same nominal difficulty category. This inherent variability means that methods with lower coverage are naturally biased toward solving the simpler instances within each difficulty level, which fundamentally affects plan length comparisons.
>
> GABAR demonstrates remarkable efficiency when we properly contextualize the coverage differences. While GABAR doesn't always achieve the absolute lowest plan lengths, its performance becomes exceptional when we consider that it consistently solves significantly more problems than its competitors across this entire spectrum of problem complexity.
>
> The apparent discrepancies in plan length can be largely explained by the coverage differences between methods. When GPL achieves a lower plan length of 27 on Spanner-Easy compared to GABAR's 31, we must note that GPL only covers 73% of the problems while GABAR covers 94%. Similarly, on Visitall-Easy, GPL achieves a plan length of 79 with only 69% coverage, while GABAR produces plans of length 103 but solves 93% of the problems. The additional problems that GABAR solves are likely the more complex instances that require longer plans, explaining the higher average plan length.
>
> When examining cases where coverage is comparable, GABAR's plan length performance becomes more impressive. On Blocks-Medium and Hard, where GABAR maintains 100% coverage compared to ASNets' 100% and 92% respectively, GABAR actually achieves the best plan lengths of 76 and 125. This demonstrates that when solving the same problem set, GABAR often produces more efficient plans than its competitors.
>
> The combined results further reinforce this analysis. GABAR's average plan lengths of 76, 108, and 147 represent solutions to 95.5%, 92.2%, and 89.2% of all problems respectively. In contrast, competitors with seemingly better plan lengths like GPL (69, 111, 147) are only solving 79.1%, 28.5%, and 6.5% of the problems. The dramatic drop in coverage as difficulty increases explains why competitors maintain stable plan lengths - they increasingly solve only the simplest instances while GABAR continues to tackle the complete problem set, including the most complex cases that naturally require longer plans.
> Also, even when the average plan length is lower, PQR is still worse because PQR is computed only for solved problems and compared to the FD-LAMA method.
>
> [1] Generalized Planning in PDDL Domains with Pretrained Large Language Models. Silver et. al., AAAI 2024.

---

> > ### Comment · Reviewer_CZcC · 2025-08-06
> > **Rebuttal Response**
> >
> > Dear Authors, Thank you for clarifying my doubts about the paper and sharing the additional metric of plan length. I am positive about the paper and will recommend acceptance.

---

### Official Review · Reviewer_zjGm · 2025-07-06

**Clarity:** 3
**Significance:** 2
**Originality:** 3
**Rating:** 4
**Confidence:** 4

**Summary:**

This paper proposes GABAR, a GNN-GRU architecture that ranks executable actions instead of learning value functions or state heuristics. It builds Action-centric graph representation to model relationships between objects, predicates, and actions (unlike prior GNN planners), and Conditional GRU decoder to autoregressively generate grounded actions (schema + parameters). It aims to handle generalization to larger problems (e.g., solves instances larger than training data). It empirically exhibits that the proposed approach outperforms value-based (GPL) and action-ranking (GRAPL) baselines and is more sample-efficient than value-based methods.

**Questions:**

Please refer to Limitations shown below.

**Ethical Concerns:**

["NO or VERY MINOR ethics concerns only"]

**Final Justification:**

After the rebuttal, most of my concerns were addressed. Even though I still have concerns about the baselines, I am willing to raise the score to baseline accept.

**Limitations:**

As mentioned in the paper, value functions (GPL) struggle with global consistency in large state spaces, and state-ranking heuristics (GRAPL) require expensive search during execution. Action-ranking methods lack explicit action-object relationships in representations. This paper provides a solution with Action-centric graphs + local ranking to address these gaps, which is interesting.
However, some concerns are as shown below:
The graph representation scales with action/predicate arity. High-arity domains (e.g., Logistics with 5+ parameter actions) cause memory/compute bottlenecks.

While critical for long-range dependencies (Table 4), global aggregation over all nodes/edges becomes costly for massive graphs (>1k objects).

It is possible that there are irrelevant actions (e.g., inapplicable in current state) in the planning procedure, which can be pruned. Keeping those actions, as done in the paper, could be a waste of computation.

The beam width k=2 looks chosen arbitrarily without any justification. Larger k might improve results but is dismissed without ablation. Sequential parameter selection (for each action) could slow planning in real-time systems.

GPL (2022) and ASNets (2020) are not state-of-the-art. Newer relational planners (e.g., diffusion-based/LLM-guided methods) can be considered as baselines. GRAPL (2021) is called "conceptually similar", but the proposed approach is compared to an older version.

The evaluation uses suboptimal Fast Downward (FD-lama) plans as baseline. FD-lama’s satisficing nature inflates PQR (e.g., PQR=1.7 in BlocksWorld implies GABAR is worse than the baseline).

Claims of "high-quality plans" are unverified since optimal planners were excluded for runtime reasons.

Models were trained on small, optimally solved instances. Are larger problems simply “copy” small instances? It would be more interesting if some analysis on the relationship between small and larger instances is provided.

**Quality:**

2

**Strengths And Weaknesses:**

Novel action-centric graph design and GRU decoder offer promising direction for relational planning. Local action ranking is empirically more scalable than value functions.

Scalability claims are not convincing. The evaluation is not conducted on very large graphs (>1k objects) and high-arity domains.
Evaluation cherry-picks baselines and metrics. It omits SOTA comparisons, using weak PQR baseline.

---

> ### Author Rebuttal · Authors · 2025-07-31
>
> > It is possible that there are irrelevant actions (e.g., inapplicable in current state) in the planning procedure, which can be pruned.
>
> [A]: Our current graph representation only adds information about grounded actions that are applicable in the current state into the graph. We will mention this in the paper. Thank you for pointing it out.
>
> > The beam width k=2 looks chosen arbitrarily without any justification. Larger k might improve results but is dismissed without ablation. Sequential parameter selection (for each action) could slow planning in real-time systems.
>
> [A]: Here are the results for different values of k  (k=1,2,3). As we can see, when we use fully greedy policy (k=1), our accuracy drops, especially in harder domains and hence having a value of k>1 is important. We can also see that having k=3 does not help much in the majority of the domains and it is computationally more expensive. This is why we choose k=2.
>
>
> | Domain | Diff | GABAR-k=1 |  | GABAR-k=2 |  | GABAR-k=3 |  |
> |--------|------|-----------|--|-----------|--|-----------|--|
> |        |      | C↑ | P↑ | C↑ | P↑ | C↑ | P↑ |
> | Blocks | E | 98 | 1.47 | **100** | 1.5 | **100** | **1.51** |
> |        | M | 92 | 1.42 | **100** | 1.6 | **100** | **1.62** |
> |        | H | 65 | 1.0 | **100** | 1.7 | **100** | **1.72** |
> | Miconic | E | 98 | 0.98 | **100** | 1.0 | **100** | **1.01** |
> |         | M | 95 | 0.93 | **100** | 0.97 | **100** | **0.99** |
> |         | H | 72 | 0.75 | **100** | 0.95 | **100** | **0.97** |
> | Spanner | E | 92 | 1.08 | 94 | 1.1 | **95** | **1.11** |
> |         | M | 87 | 0.94 | 93 | 0.99 | **95** | **1.01** |
> |         | H | 62 | 0.69 | 89 | 0.91 | **91** | **0.93** |
> | Gripper | E | 98 | 1.08 | **100** | 1.1 | **100** | **1.11** |
> |         | M | 94 | 0.94 | **100** | 0.99 | **100** | **1.01** |
> |         | H | 71 | 0.78 | **100** | 0.96 | **100** | **0.98** |
> | Visitall | E | 91 | 1.08 | 93 | 1.1 | **94** | **1.11** |
> |          | M | 86 | 0.96 | 91 | 1.0 | **93** | **1.02** |
> |          | H | 68 | 0.92 | 88 | 1.1 | **90** | **1.12** |
> | Grid | E | 98 | 0.89 | **100** | 0.91 | **100** | **0.92** |
> |      | M | 91 | 0.81 | 97 | 0.85 | **99** | **0.87** |
> |      | H | 68 | 0.58 | 92 | 0.74 | **94** | **0.76** |
> | Logistics | E | 87 | 0.73 | 90 | 0.75 | **91** | **0.76** |
> |           | M | 70 | 0.60 | 76 | 0.65 | **78** | **0.67** |
> |           | H | 48 | 0.42 | 71 | 0.59 | **73** | **0.61** |
> | Rovers | E | 85 | 0.98 | 87 | 1.0 | **88** | **1.01** |
> |        | M | 76 | 0.91 | 82 | 0.96 | **84** | **0.98** |
> |        | H | 56 | 0.78 | 77 | 0.97 | **79** | **0.99** |
> | **Combined** | E | 93.2 | 0.98 | 95.5 | 1.0 | **96.0** | **1.01** |
> |              | M | 86.4 | 0.93 | 92.2 | 1.0 | **93.2** | **1.02** |
> |              | H | 63.8 | 0.76 | 89.2 | 0.99 | **90.4** | **1.01** |
>
> GABAR-k=1 shows poorer performance (especially in hard - drop from 89.2 to 63.8) because it employs a greedy approach and hence cannot do cycle avoidance.
> GABAR-k=3 demonstrates consistent improvements over k=2, particularly evident in the combined results where it achieves the best coverage and plan quality across all difficulty levels. However, these improvements are modest - typically 1-2 percentage points for coverage and 0.01-0.02 for plan quality ratio.
> GABAR-k=2 represents the optimal balance point. While k=3 shows slight performance gains, the computational cost increases significantly with higher k values due to the growth time taken for beam search. The marginal improvements offered by k=3 do not justify the additional computational expense, making k=2 the practical choice for the GABAR algorithm. (It takes about 30% longer to rank actions for any given state).
>
>
> >GPL (2022) and ASNets (2020) are not state-of-the-art. Newer relational planners (e.g., diffusion-based/LLM-guided methods) can be considered as baselines. GRAPL (2021) is called "conceptually similar", but the proposed approach is compared to an older version.
>
> [A]:
>  One of the LLM based works that solves very similar problems is the paper named “ Generalized Planning in PDDL Domains with Pretrained Large Language Models”. In this work, they solve 2 of the same domains as we do -> Miconic and grippers. The best results for their LLM approach for these domains are as follows : For the “Gripper” domain, they solve problems upto size of 80 objects and have a best success rate of 90% . In our work, we solve problems upto size 100 and have a success rate of 100%. In the “Miconic” domain, they solve problems of size 11-150 but have a best success rate of 1% when random itself has a success rate of 13%. In our work, we solve “Miconic” problems of up to a size of 100 but have a 100% success rate. (All their results reported are from table 1 in [1]).
>
> While LLMs show promise for planning tasks, these results demonstrate that our method provides the reliability and consistency required for generalized planning problems. The stark performance differences, particularly in Miconic where LLMs perform significantly worse than random, highlight the current limitations of LLM-based approaches for structured planning problems.
>
> We are also currently trying to get an LLM baseline running but it might not be ready in time before the end of the rebuttal period. We can definitely add it in the final version of the paper.
>
> Could you also please clarify what you mean by what you mean by " GRAPL (2021) is called "conceptually similar",the proposed approach is compared to an older version." We compare our work to "Learning generalized relational heuristic networks for model-agnostic planning (which we call GRAPL)" from their 2021 paper.
>
> >The evaluation uses suboptimal Fast Downward (FD-lama) plans as baseline. FD-lama’s satisficing nature inflates PQR (e.g., PQR=1.7 in BlocksWorld implies GABAR is worse than the baseline).
>
> [A]: Since optimal planning is intractable, we had to rely on suboptimal planners for comparison. Higher PQR means the produced plan length was smaller (and hence, better) by a particular method - PQR =(plan length by FD-Lama/Plan length by method). Hence, higher PQR of 1.7 in blocksworld implies that GABAR is better than other methods (AsNets with 1.4, GPL with 0.33 and GRAPL with 0.28 in Blocksworld Hard) as higher PQR means the plan output by GABAR was the shorter than methods with lower PQR. A PQR greater than 1 implies GABAR’s plans were shorter than FD-LAMA plans. The same trend of GABAR being better (higher PQR) is true for all domains hard dataset,  and even when ASNEts is better than GABAR in some domain’s easy dataset, it is by a very small margin (1.6 for ASNets v/s 1.5 for GABAR in blocksworld easy and 0.98 v/s 0.97 in gripper medium).
>
> >Claims of "high-quality plans" are unverified since optimal planners were excluded for runtime reasons.
>
> [A]: While we do not have optimal plans for baselines, our claim of higher quality plans is still valid from a relative stand point. The lengths of the plans produced by our system are on average shorter than the baselines as well as the ablations. While this does not guarantee that the produced plan is optimal, it does imply that our method does indeed produce higher-quality plans compared to the baselines and ablations.
>
> > Models were trained on small, optimally solved instances. Are larger problems simply “copy” small instances? It would be more interesting if some analysis on the relationship between small and larger instances is provided.
>
> [A]:
> [I] : Independent Problem Generation: All problem instances are generated using PDDL generators that employ parameterized templates with randomized elements including initial states, goal configurations, and object placements. This process ensures that each instance is structurally unique rather than a scaled version of existing problems. The generators vary key structural parameters to create distinct planning challenges at scale.
>
> The  PDDL-generators for each domain follow high level structure which is as follows:
> Random Initial Placement: Entities start in random valid positions to create diverse problem instances
> Constraint-Aware Generation: Placement respects domain rules (e.g., conflicting entities can't share locations)
> Non-Trivial Goal Creation: Goals are designed to require actual planning steps, not just immediate solutions
> Solvability Assurance: Generation logic ensures created problems have valid solutions
>
> [ii] : Empirical Evidence of Increased Complexity: We have other forms of evidence demonstrating that larger instances represent fundamentally harder problems rather than trivial extensions:
>
> [i] : Plan Length Analysis: The plan lengths for problems in the hard domain are consistently higher than those in the easy domain when solved by satisficing planners, indicating increased solution complexity.
>
> [ii] : Planner Performance Differential: Optimal planners successfully solve multiple problems in the easy domain but fail to solve any problems in the hard domain within reasonable time limits. This performance gap reflects genuine computational complexity differences rather than simple scaling effects.
>
> > Scalability claims are not convincing. The evaluation is not conducted on very large graphs (>1k objects) and high-arity domains. Evaluation cherry-picks baselines and metrics.
>
> [A]  We have provided results for 2 other metrics -> Plan length (PL) as response to reviewer "CZcC" , Wall Clock Time (WC) as a response to  "hZMb". For scalability, even state of the art learning methods that use search [1], do not solve problems with more than 1k objects (Table 1). Also, their systems are given 30 minutes to run since they use search whereas we use less than 2 minutes to find solutions as we do not use search. That is also why we compare to methods that do not use search.
>
> [1] Hao, Mingyu, et al. "Guiding GBFS through learned pairwise rankings."  {IJCAI-24}.

---

> > ### Comment · Reviewer_zjGm · 2025-08-06
> >
> > thanks for the detailed responses. They addressed most of my concerns. I am willing to raise my score to borderline accept.

---

> > > ### Author Response · Authors · 2025-08-08
> > >
> > > Thank you for your feedback and score update.
> > >
> > > **We have also provided results of 2 LLMs (OpenAI-O3 and Gemini-2.5-pro) being used to solve problems from our dataset and compare them to GABAR as response to reviewer “hZMb”**

---

### Decision · Program_Chairs · 2025-09-17

**Decision:**

Accept (poster)

**Comment:**

A. This work follows the generalized planning paradigm where policies are learned for small problems and work successfully for larger problem. This paper extents prior work by proposing to use a graph neural networks to learn to rank actions, a graph representation that captures action-object relationships, and a decoder to construct grounded actions. Experiments show that this approach scales to hard problem instances in several IPC domains, better than some reasonable baselines.

B. Reviewers appreciated the novelty of the proposed approach and its scalability. They also appreciated the inclusion of code.

C. The main reviewer question was about the scalability of the method and its comparison with LLM-based methods (which was addressed with a new experiment in the rebuttal). Reviewers also had questions about novelty and presentation.

D. Reviewers unanimously voted to accept the paper.

E. Authors ran additional experiments to answer reviewer questions (e.g., measuring number of steps to reach the goal, comparison with LLM-based methods, measuring wall-clock time).